



# Free amino acids quantification in cloud water at the puy de
# Dôme station (France)
Pascal Renard[1], Maxence Brissy[2], Florent Rossi[2], Martin Leremboure[2], Saly Jaber[2], Jean-Luc
Baray[1,3], Angelica Bianco[1], Anne-Marie Delort[2,*] and Laurent Deguillaume[1,3,*]
[1] Université Clermont Auvergne, Laboratoire de Météorologie Physique, OPGC/CNRS UMR 6016, Clermont-Ferrand,
France.
[2] Université Clermont Auvergne, CNRS, SIGMA Clermont, Institut de Chimie de Clermont-Ferrand (ICCF), Clermont-
Ferrand, France.
[3] Université Clermont Auvergne, Observatoire de Physique du Globe de Clermont-Ferrand, UAR 833, Clermont-Ferrand,
France.
*Correspondence to*: Anne-Marie Delort (a-marie.delort@uca.fr) and Laurent Deguillaume (laurent.deguillaume@uca.fr)
**Abstract.**
Eighteen free amino acids (FAAs) were quantified in cloud water sampled at the puy de Dôme station (PUY - France)
during 13 cloud events. This quantification has been performed without concentration neither derivatization, using LC-
MS and the standard addition method to avoid matrix effects. Total concentrations of FAAs (TCAAs) vary from 1.2 μM
to 7.7 μM, Ser (Serine) being the most abundant AA (23.7 % in average) but with elevated standard deviation, followed
by Glycine (Gly) (20.5 %), Alanine (Ala) (11.9 %), Asparagine (Asn) (8.7 %), and Leucine/Isoleucine (Leu/I) (6.4 %).
The distribution of AAs among the cloud events reveals high variability. TCAA constitutes between 0.5 and 4.4 % of the
dissolved organic carbon measured in the cloud samples. AAs quantification in cloud water is scarce but the results agree
with the few studies that investigated AAs in this aqueous medium. The environmental variability is assessed through a
statistical analysis. This work shows that AAs are correlated with the time spent by the air masses in the boundary layer,
especially over the sea surface before reaching the PUY. The cloud microphysical properties fluctuation does not explain
the AAs variability in our samples confirming previous studies at PUY. We finally assessed the sources and the
atmospheric processes that potentially explain the prevailing presence of certain AAs in the cloud samples. The initial
relative distribution of AAs in biological matrices (proteins extracted from bacterial cells or mammalian cells, for example)
could explain the dominance of Ala, Gly and Leu/I. AA composition of aquatic organisms (*i.e.*, diatoms species) could
also explain the high concentrations of Ser in our samples. The analysis of the AAs hydropathy also indicates a higher
contribution of AAs (80 % in average) that are hydrophilic or neutral revealing the fact that other AAs (hydrophobic) are
less favorably incorporated into cloud droplets. Finally, the atmospheric aging of AAs has been evaluated by calculating
atmospheric lifetimes considering their potential transformation in the cloud medium by biotic or abiotic (mainly
oxidation) processes. The most concentrated AAs encountered in our samples present the longest atmospheric lifetimes
and the less dominant are clearly efficiently transformed in the atmosphere, potentially explaining their low concentrations.
However, this cannot fully explain the relative contribution of several AAs in the cloud samples. This reveals the high
complexity of the bio-physico-chemical processes occurring in the multiphasic atmospheric environment.
## 1. Introduction
Free or combined amino acids (AAs) that make up proteins and cell walls in living organisms are ubiquitous chemical
compounds found in various environments. In the atmosphere, they are commonly detected in the condensed phases due


to their low vapor pressures. They have been studied and characterized in atmospheric particles (Barbaro et al., 2020;
Matos et al., 2016), rain water (Mace et al., 2003a; Mace et al., 2003b; Xu et al., 2019; Yan et al., 2015), fog water (Zhang
and Anastasio, 2003b) and more recently in cloud water (Bianco et al., 2016b; Triesch et al., 2021). Many efforts have
been made in the past to assess their sources, their role in the atmospheric chemical and physical processes and their fate
(Cape et al., 2011). However, despite those investigations, their exact role in the atmosphere is still poorly understood.
They have been studied for their hygroscopic properties since they can modify the ability of the particle to act as cloud
condensation nuclei (CCN) (Chan et al., 2005; Kristensson et al., 2010; Li et al., 2013) or ice nuclei (IN) (Pummer et al.,
2015; Szyrmer and Zawadzki, 1997). More recently, the role of AAs in new particle formation has been also discussed
(Ge et al., 2018). This raises the question of their role in aerosol and cloud formation and hence in the radiative forcing
of the Earth's surface. In atmospheric aqueous phases, some AAs have been found to potentially influence atmospheric
chemistry by reacting with atmospheric oxidants (Bianco et al., 2016b; McGregor and Anastasio, 2001; Zhang and
Anastasio, 2003a); the study from De Hann et al. even showed that AAs can react with glyoxal to form secondary aerosol
mass (De Haan et al., 2011). AAs are part of the proteinaceous fraction of aerosol particle that significantly contribute to
the organic carbon and organic nitrogen fraction of aerosol particles. Their presence in aerosol particles can modify their
chemical properties such as acidity/basicity and buffering ability (Cape et al., 2011; Zhang and Anastasio, 2003b). Finally,
AAs are also transferred by atmospheric deposition to other ecosystems such as aquatic surfaces where they act as
nutrients since they are particularly bioavailable (Wedyan and Preston, 2008). Atmospheric AAs can therefore contribute
to the nutrient cycling at global scale as well as the global carbon and nitrogen cycles.
AAs have been detected in the atmosphere under various contrasted environmental scenarios such as urban area (Barbaro
et al., 2011; Di Filippo et al., 2014; Ren et al., 2018; Zhu et al., 2020), background/rural sites (Bianco et al., 2016b; Helin
et al., 2017; Samy et al., 2011; Song et al., 2017), marine environment (Mandalakis et al., 2011; Matsumoto and Uematsu,
2005; Triesch et al., 2021; Violaki and Mihalopoulos, 2010) and polar regions (Barbaro et al., 2015; Feltracco et al., 2019;
Mashayekhy Rad et al., 2019; Scalabrin et al., 2012). The quantity and type of AAs detected in all the compartments
(aerosol particles, cloud water, rainwater) vary over a wide range. Indeed, their emissions, residence times and spatial and
temporal distributions are driven by complex bio-physico-chemical processes occurring in the atmosphere (transport,
chemical and biological transformations, deposition, *etc.*). Proteinaceous materials detected in the atmosphere are in
majority linked with emissions of primary biological aerosol particles that notably include viruses, bacteria, fungi, algae,
spores, pollens and fragments of plants and insects (Després et al., 2012; Fröhlich-Nowoisky et al., 2016). The main
source is consequently from biogenic origin, but several anthropogenic sources can also contribute (industry, agricultural
practices, wastewater treatment). It is suggested that AAs are directly emitted into the atmosphere or result from the
transformations of proteins by enzymatic activity, decomposition by the temperature or the photochemistry (Mopper and
Zika, 1987). There are some studies highlighting other possible sources such as emissions by volcanoes (Scalabrin et al.,
2012), biomass burning emissions (Chan et al., 2005) and marine emissions by sea bubble bursting (Barbaro et al., 2015;
Matsumoto and Uematsu, 2005). Due to the wide variety of AAs sources in the atmosphere, it is rather difficult to correlate
AA concentration and speciation with specific sources: Abe et al. (2016) recently proposed to use AAs as markers for
biological sources in urban aerosols (Abe et al., 2016); Matsumoto and Uematsu (2005) suggested that the major source
of free amino acids (FAAs) in aerosols over the remote North Pacific are related to long-range transport from continental
areas; Scalabrin et al. (2012) used AAs ratio to evaluate aerosol aging in the atmosphere.
The analysis of AAs in the atmosphere is essential and has been widely conducted to document the concentrations of
aerosol particles, their environmental variability, and their effects on atmospheric physico-chemical processes. AAs can



also be transferred into the atmospheric aqueous media after activation of aerosol particles into cloud droplets. They
consequently contribute to the complex dissolved organic matter measured in clouds that is composed by a significant
fraction of biological-derived material (lipids, peptides, carbohydrates…) (Bianco et al., 2018; Cook et al., 2017; Zhao et
al., 2013). However, only few studies focus on the detection of AAs in cloud water (Bianco et al., 2016b; Triesch et al.,
2021) mainly because of the inherent difficulty to sample clouds. AA concentration in cloud water results from the
dissolution of the soluble fraction of the aerosol particles acting as CCN and IN; some very recent studies also argue that
AAs could be processed in the cloud medium by the biological activity (Bianco et al., 2019). For instance, the
biodegradation of AAs was demonstrated to occur in rainwater (Xu et al., 2020) and in microcosms mimicking the cloud
environment (Jaber et al., 2021). The presence of transcripts of genes coding for AAs and proteins biosynthesis and
biodegradation has been also shown directly in cloud water samples (Amato et al., 2019). AAs can also be photo-
transformed by abiotic processes mainly implying oxidants (Jaber et al., 2021). They can produce other compounds such
as carboxylic acids, nitrate, and ammonia (Berger et al., 1999; Berto et al., 2016; Bianco et al., 2016a; Marion et al., 2018;
Pattison et al., 2012), thus potentially contributing to the formation in aqueous phase of secondary organic aerosol
("aqSOA"). It is therefore crucial to document AA concentration levels and speciation in clouds.
This aim of this work is devoted to the quantification of FAAs in cloud waters. This is quite a challenge due to the
chemical complexity of the cloud medium and the low concentration of FAAs ($\approx \mu M$). In atmospheric waters, namely
fog (Zhang and Anastasio, 2003b), rain (Gorzelska et al., 1992; Mopper and Zika, 1987; Xu et al., 2019; Yan et al., 2015)
and clouds (Bianco et al., 2016b), the main technique that has been commonly used is liquid chromatography coupled
with fluorescence detection. This approach is based on pre- or post-column derivatization of the AAs to increase the
sensitivity and simplify the separation by chromatography, but it is time consuming. More recently, Triesch et al. (2021)
have used liquid chromatography hyphened to mass spectrometry (LC-MS) to detect derivatized AAs after concentration
of cloud water samples. The use of LC-MS represents a significant improvement as it allows a unique identification. We
propose here to go further using LC-MS without pre-concentration and derivatization of the sample. In addition, to avoid
matrix effect, we propose to quantify the AAs by the standard addition method (Hewavitharana et al., 2018). Cloud
sampling is performed at the puy de Dôme station (PUY) in France offering possibility to collect 13 samples for various
environmental conditions. Variability of cloud AA concentrations together with cloud bio-physico-chemical properties
and air mass history is thus discussed in this work.
## 2. Methods/Materials
### 2.1 Site and cloud sampling
13 clouds were sampled from 2014 to 2020 at the puy de Dôme station (PUY) in France (45.77 °N, 2.96 °E; 1465m a.s.l.).
This mountain observatory is part of the multi-site platform CO-PDD combining *in situ* and remote sensing observations
at different altitudes (Baray et al., 2020). PUY belongs to international atmospheric survey networks: ACTRIS (Aerosols,
Clouds, and Trace Gases Research Infrastructure), EMEP (the European Monitoring and Evaluation Program) and GAW
(Global Atmosphere Watch) as example. Meteorological parameters, atmospheric gases, aerosols, and clouds are
monitored over long-term period, to investigate the bio-physico-chemical processes linking those elements and to evaluate
the anthropogenic forcing on climate.
The sampling is performed using cloud water collectors previously described (Deguillaume et al., 2014), under non-
precipitating and non-freezing conditions. Before cloud collection, the cloud impactors are cleaned and sterilized by





autoclaving. After sampling, the cloud waters are stored in sterilized bottles; a fraction of the aqueous volume is filtered

using a 0.2 μm nylon filter within 10 min after sampling to eliminate non soluble particles and microorganisms. The

samples are then kept in the dark and stored at 4°C or kept at -80°C (depending on the targeted compounds) before the

analyses.

## 2.2  Physical, chemical, and microbiological characterization of clouds

A systematic characterization is performed on cloud samples allowing to document the available PUYCLOUD database

(http://opgc.fr/vobs/so_interface.php?so=puycloud) of the cloud water chemical and biological composition (Renard et

al., 2020). These data are reported in Table S1 for the studied cloud events.

Chemical composition analyses are performed on cloud samples: pH, total organic carbon (TOC) concentration, and

concentrations of main inorganic ionic species. TOC analyses are performed with a TOC analyzer (Shimadzu). The

spectrofluorimetric method based on the reactivity of p-hydroxyphenilacetic acid with horseradish peroxidase is used to

measure the concentration of $H_2O_2$ in cloud water (Wirgot et al., 2017). Ionic inorganic species ($Ca^{2+}$, $K^+$, $Mg^{2+}$, $Na^+$,

$NH_4^+$, $Cl^-$, $SO_4^{2-}$ and $NO_3^-$) are measured by ion chromatography (Deguillaume et al., 2014).

Cloud microphysical properties are determined with the Gerber particle volume monitor-100 (PVM-100) providing liquid

water content (LWC) and effective droplet radius ($r_e$) parameters.

The biology of cloud water is also assessed by quantification of bacteria density (CFU $mL^{-1}$) at 17°C (Vaïtilingom et al.,

2012) and adenosine triphosphate (ATP) concentration is measured using the BioThema© ATP Biomass kit HS (Koutny

et al., 2006).

## 2.3  Quantification of Amino Acids (AAs)

### 2.3.1   Sample preparation

Before analysis by LC-MS, in order to apply the standard addition method to quantify AAs (Hewavitharana et al., 2018),

standard solutions are used to spike cloud water samples. Standard solutions are prepared in ultra-pure water and

contained alanine (Ala, SIGMA-ALDRICH), arginine (Arg, SIMAFEX), asparagine (Asn, SIGMA-ALDRICH),

aspartate (Asp, SIGMA-ALDRICH), glutamine (Gln, SIGMA-ALDRICH) glutamic acid (Glu, SIGMA-ALDRICH),

glycine (Gly, MERCK), histidine (His, SIGMA-ALDRICH), leucine/isoleucine (Leu/I, SIGMA-ALDRICH), lysine (Lys,

SIGMA-ALDRICH), methionine (Met, SIGMA-ALDRICH), phenylalanine (Phe, ACROS organics), proline (Pro,

SIGMA-ALDRICH), serine (Ser, SIGMA-ALDRICH), threonine (Thr, SIGMA-ALDRICH), tryptophan (Trp, SIGMA-

ALDRICH), tyrosine (Tyr, SIGMA-ALDRICH), valine (Val, SIGMA-ALDRICH), cysteine (Cys, SIGMA-ALDRICH).

The ratio between the sample volume and the standard solution volume is respectively (9:1). The mixture is then vortex

mixed for 1 min.

Ten samples ready for LC-MS analysis are prepared containing the original cloud water added with 20 AAs at final

concentrations set to 1.0, 5.0, 10, 25, 50, 100, 150, 500 μg $L^{-1}$. This range of concentrations is appropriate considering

previous quantification of AAs in cloud waters sampled at PUY (Bianco et al., 2016b). This also allows to cover large

range of AA concentrations that can be highly variable depending on the cloud events. L-Lys isotope ($^{13}C_6$, 99 %; $^{15}N_2$,

99 %) is also added to each sample at the concentration of 15 μg $L^{-1}$ for mass calibration (m/z = 155.1273).





### 2.3.2 LC-MS analysis

LC-MS analyzes are performed using an UltiMate™ 3000 (Thermo Scientific™) LC equipped with a Q-Exactive™ Hybrid Quadrupole-Orbitrap™ Mass Spectrometer (Thermo Scientific™) ionization chamber. Chromatographic separation of the analytes is performed on BEH Amide/HILIC (1.7 μm, 100 mm × 2.1 mm) column with column temperature of 30°C. The mobile phases consist of 0.1 % formic acid and water (A) and 0.1% formic acid and acetonitrile (B) with a 0.4 mL min$^{-1}$ flow rate. A four-step linear gradient is applied during the analysis: 10 % A and 90 % B in 8 min, 42 % A and 58% B in 0.1 min, 50 % A and 50 % B for 0.9 min and 10 % A and 90 % B for 3 min.

The Q-Exactive ion source is equipped with electrospray ionization (ESI) and the Q-Orbitrap™. The volume of injection is 5 μL, and the flow injection analyses are performed for individual AA solutions to obtain the mass spectral data, from which ions are carefully chosen for analysis in the selected ion monitoring (SIM) mode, using the above-mentioned parameter conditions. The mass resolution is set to 35000 FWHM (full width at half maximum), and the instrument is tuned for maximum ion throughput. AGC (automatic gain control) target or the number of ions to fill C-Trap is set to 10$^5$ with injection time of 100 ms. Tests with standard solution and cloud water samples show a better sensitivity in positive mode of ionization for all AAs and with a preference for [M+H]$^+$ ionization (ESI+). Other Q-Exactive$^{TM}$ generic parameters are: N$_2$ flow rate set at 13 a.u, sheath gas (N$_2$) flow rate set at 50 a.u, sweep gas flow rate set at 2 a.u, spray voltage set at 3.2 kV in positive mode, capillary temperature set at 320°C, and heater temperature set at 425°C.

Analysis and visualization of the data set are performed using Xcalibur™ 2.2 software; it allows controlling and processing data from Thermo Scientific™ LC-MS systems and associated instruments. Examples of chromatograms and MS spectra for three AAs (Ser, Val and Trp) are presented in Figures S1 (a, b, c). For quality control, one cloud sample has been analyzed in MS² to check the presence of isobaric molecules. The peak with a retention time of 2.89 min and m/z of 118,0867 [M+H]$^+$ has been found to correspond to the mixture of 2 isobaric molecules: valine and betaine (Figure S2). Therefore, Val cannot be quantified. Leu and Ile could not be also distinguished as there are isobaric with the same retention time (hereafter, Leu/I). Cys is not quantifiable as it forms S-S bonds. Consequently, 18 AAs can be quantified in this study: Ala, Arg, Asn, Asp, Gln, Glu, Gly, His, Leu/I, Lys, Met, Phe, Pro, Ser, Thr, Trp, Tyr. The retention times and exact masses measured by LC-MS of all the AAs are summarized in Table S2.

### 2.3.3 Standard addition

Cloud water is a complex mixture, conducive to disturbance in the LC-MS analytical signal. To restrain this matrix effect, the AAs quantification is performed with the method of the standard addition, which consists of the addition of a series of small volumes of concentrated standard to an existing unknown. For each AA, this method provides a calibration curve. Figure S3 presents, as an example, how the concentration of Gly is measured for a particular cloud event (11 Mar cloud) using the standard addition method. The magnitude of the intercept on the x-axis of the trendline is the original concentration of Gly.

Table S3 displays calibration curve data measured for the 13 different cloud samples for each AA. The linearity of the calibration curves is attested by the high R$^2$ values (> 0.95). The AA concentrations and their standard deviation (STD) are calculated according to the equation from Broekaert and Daniel (Broekaert, 2015). More details can be found in SI (Figure S3 and attached explanations of the calculations).



### 2.4 Evaluation of air mass history

The CAT model (Computing Atmospheric Trajectory Tool) is a three-dimensional (3D) forward/backward kinematic trajectory code which has been recently developed and used to characterize the atmospheric transport of air masses reaching PUY station (Renard et al., 2020). Backward trajectories clusters have been calculated for all clouds included in the PUYCLOUD database. The temporal resolution of the backtrajectories is 15 min and the total duration is 72 h. The model is initialized with wind fields from the ECMWF ERA-5 reanalyzed with a horizontal resolution of 0.5° and 23 vertical pressure levels between 200 and 1000 hPa. On the basis of the atmospheric boundary layer height (ABLH) and altitude of topography interpolated for each trajectory point, this numerical tool allows to calculate the percentage of points above the sea and the continental surfaces (Sea surface *vs* Continental surface), hereafter named "zone". A "zone matrix" is thus constructed from CAT model outputs and used for a statistical classification of each cloud event. All the data relative to the 13 clouds of this study are reported in Table S1.

This classification proposed by Renard et al. (2020) is based on a statistical analysis considering the cloud chemical concentrations of the PUYCLOUD dataset. This allows clustering clouds in four categories: "Highly marine", "Marine", "Continental" and "Polluted". The "Marine" clouds come predominantly from western sectors and have the lowest ion concentrations. Marine category is predominant and the most "homogeneous" in terms of concentrations. The "Highly marine" category with a similar air mass history, gathers the clouds with the highest sea-salts concentrations ($Cl^-$, $Mg^{2+}$ and $Na^+$). The Continental category corresponds mainly to air masses arriving from the northeast sector with high concentrations of potentially anthropogenic ions ($NH_4^+$, $NO_3^-$ and $SO_4^{2-}$). Finally, the "Polluted" category gathers some cloud samples with the highest anthropogenic ion concentrations. All the data relative to the clouds studied in the present work are reported in Table S1.

### 2.5 Statistical analysis

With the objective to categorize cloud samples, we performed agglomerative hierarchical clustering (AHC), an iterative classification, based on AA concentrations. The AHC dendrogram shows the progressive grouping of the data. The dissimilarity between samples is calculated with the Ward's agglomeration method using Euclidean distance. The number of categories to retain is automatically defined on the base of the entropy (Addinsoft, 2020).

A large variability of the AA concentrations and relative proportions in the 13 cloud samples from PUY is observed. In order to better understand this variability, a PLS regression is performed to analyze the correlations between the explanatory (X) and dependent (Y) variables. The Xs variables gather the biological (ATP and bacteria density), physical (temperature and pH) and chemical (TOC, $Ca^{2+}$, $K^+$, $Mg^{2+}$, $Na^+$, $NH_4^+$, $Cl^-$, $NO_3^-$ and $SO_4^{2-}$ concentrations) parameters, the "zone" matrix (Sea/Continental surface </> ABLH), as well as the seasons. The Ys variables are the 18 AA concentrations (Ala, Arg, Asn, Asp, Gln, Glu, Gly, His, Leu/I, Lys, Met, Phe, Pro, Ser, Thr, Trp, Tyr). The Mann–Whitney nonparametric tests are carried out to validate significant differences (p-value < 0.05) between two groups (Renard et al., 2020).

## 3. Results

### 3.1 Evaluation of LC-MS technique for a direct measurement of AAs in cloud

The analytical method used in this study allows assaying AAs directly in cloud samples. MS hyphenated to LC allows the analysis of the underivatized and non-concentrated analytes, avoiding potential biases and time-consuming processes. The standard addition method also prevents matrix effects which are very commonly encountered with environmental



matrices (Hewavitharana et al., 2018). 18 AAs in cloud water sampled at PUY have been identified and their
concentrations quantified (Table S1). Concentrations and standard deviation (STD) values obtained for all AAs and cloud
samples are reported in Table S3 and detailed in Figure S3. The median STD is 12 nM (ranging from 6 nM for Trp to 44
nM for Ser). The relative standard deviation (RSD) ranges from 8 % for Ala to 119 % for Arg (median: 23 %).
The STD values, as calculated in this work in the context of a standard addition (equation detailed in Figure S3), could
be compared to the limit of quantification (LOQ) established in works using internal standard method (Broekaert, 2015).
Both equations are similar and provide comparable results. However, the precision depends on the number of standard
points added in the method, and not on the number of replicates. The values in this work are globally low and consistent
with those reported in previous works on cloud waters and aerosol particles (Table S4). A recent study performed by
Triesch et al. (2021) was able to quantify Val in cloud water samples, but they could not measure Arg, Asn, His, Lys, Cys
and Tyr concentrations. Triesch et al. (2021) is also based on LC-MS (Orbitrap™), but with samples concentrated (factor
44) and derivatized with a pre-column. They reported LOQ values ranging from 0.2 to 1.0 µg L$^{-1}$, *vs* STD from 1.1 to 4.6
µg L$^{-1}$ in this study. STDs are also within the same range of magnitude than those reported on aerosol particles by Helin
et al. (2017) using direct injection of extracted AAs in LC-MS (triple-quadrupole technology), with values varying from
4 to 160 nM, *vs* STD values from 8 to 44 nM in this work.
Looking more carefully at the median of the AA concentrations RSDs (calculated from data displayed in Table S3), it
appears that some AAs (Ala, Gly, Leu/I, Pro, Ser and Thr) have low RSDs (from 8 to 13 %) while others (Tyr, Lys, Trp,
Gln, Met and Arg) present higher RSDs (from 44 to 119 %). The RSD values obtained in this work are within the same
range of order than those reported by Helin et al. (2017). To conclude, these uncertainties do not change the final range
of magnitude of the AA concentrations.
**3.2 Cloud physico-chemical characteristics**
Table S1 presents data characterizing properties of cloud samples (chemical composition, microphysical properties, air
mass history). Among the 13 studied clouds, 12 clouds are classified as "Marine" according to their ion concentrations
(Renard et al. 2020). 17 Jul cloud from the North-East is classified as "Continental" due to its $NH_4^+$, $NO_3^-$ and $SO_4^{2-}$
concentrations that are significantly higher than the other studied clouds (Table S1).





Similarly to the work performed by Renard et al. (2020), the CAT model is used to characterize the air mass history of
the cloud samples. Figure 1 represents the mean backtrajectories calculated over the sampling period of the 13 cloud
samples; Figure S4 presents the backtrajectory calculations, every hour, for individual cloud event over the sampling
period. The CAT model provides a "zone matrix" (Table S1) gathering the percentage of time spent by the air masses
over the Sea surface and over the Continental surface with the discrepancy between the presence in the boundary layer (<
ABLH) or in the free troposphere (> ABLH). During the 72h backtrajectories, the air masses, in average, spent a
significant time in free troposphere ($\approx 80\%$), and above the Sea surface (56 %) (Figure 1 and Table S1). This is consistent
with the conclusion from Renard et al. (2020) arguing that the marine category is the most encountered one at PUY; a
category characterized by a low ionic content. However, even if the sampled clouds belong mainly to one category
("Marine"), they present chemical compositions that vary significantly from one sample to the other. This is discussed in
the following section where AAs content is presented, and its variability analyzed.

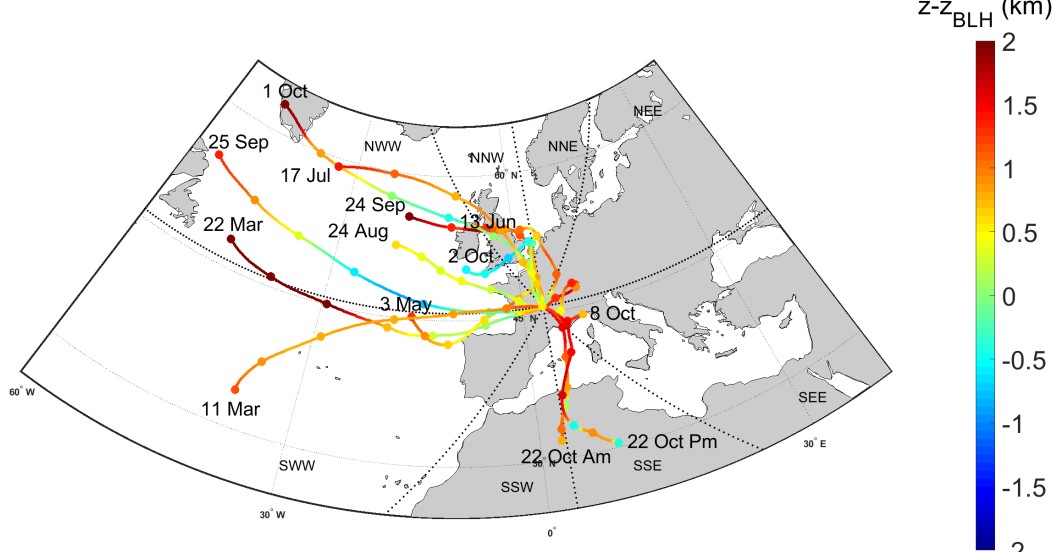


**Figure 1. Backtrajectory plots of air masses reaching PUY. Colors correspond to the air mass height minus the atmospheric**
**boundary layer height (ABLH). Positive values (> ABLH, red) indicate the air mass is in the free troposphere. Negative values**
**(< ABLH, blue) indicate the air mass is in the boundary layer. Each trajectory plot is the mean value of a cluster of 45 CAT**
**trajectories calculated over 72 h, every hour from the begin to the end of the cloud sampling period. Trajectory points are**
**calculated every 15 min and dots on the figure indicate 12 h intervals. All the trajectory clusters (without averaging) for each**
**of the 13 events are given in Figure S4.**
**3.3  Quantification of AAs in cloud waters**
**3.3.1    Concentration and distribution of AAs at PUY**
AA concentrations (in nM) measured in the 13 cloud samples are reported in Table S1. Figure 2 represents the distribution
of AA concentrations; minimum, maximum, mean, STD, and RSD of concentrations of those compounds are reported in
Table 1.






**Table 1. Distribution of AA concentrations measured in the 13 clouds sampled at PUY: minimum, maximum, mean, standard**
**deviation (STD), and relative standard deviation (RSD).**

| Label | Minimum (nM) | Maximum (nM) | Mean (nM) | σ (nM) | RSD |
|---|---|---|---|---|---|
| Ser | 4 | 2983 | 721 | 866 | 120 % |
| Gly | 123 | 1787 | 622 | 507 | 81 % |
| Ala | 96 | 862 | 360 | 270 | 75 % |
| Asn | 8 | 1105 | 264 | 375 | 142 % |
| Leu/I | 60 | 577 | 194 | 141 | 72 % |
| Thr | 23 | 462 | 176 | 133 | 75 % |
| Asp | 33 | 543 | 165 | 166 | 100 % |
| Pro | 76 | 290 | 137 | 72 | 53 % |
| Glu | 6 | 244 | 87 | 70 | 81 % |
| His | 16 | 185 | 65 | 61 | 93 % |
| Phe | 6 | 133 | 57 | 39 | 68 % |
| Tyr | 13 | 165 | 55 | 50 | 91 % |
| Lys | 0 | 141 | 50 | 48 | 96 % |
| Gln | 2 | 111 | 33 | 36 | 108 % |
| Arg | 4 | 52 | 25 | 17 | 69 % |
| Trp | 3 | 26 | 14 | 9 | 66 % |
| Met | 3 | 27 | 11 | 13 | 119 % |
| **TCAA** | **1187** | **7749** | **2696** | **1936** | **72 %** |

The total concentrations of free amino acids (TCAAs) vary significantly between cloud samples: the lowest concentration
is 1.2 µM (24 Sep cloud), and the highest one is 7.7 µM (2 Oct cloud) while the mean value is equal to 2.7 µM (Table 1
and Figure 3a). In detail, Ser is the most abundant AA in the 13 cloud samples, with the highest STD (from 4 to 2983
nM), followed by Gly (from 123 to 1787 nM), Ala (from 96 to 862 nM), and Asn (from 8 to 1105 nM) (Figure 2). This
ranking seems common and ubiquitous, from polar to urban sites, in clouds as in rainwaters or aerosols (Table S4). Ser
is also preponderant in marine clouds at Cabo Verde Islands (Triesh et al., 2021) and rural fogs in Northern California
(Zhang and Anastasio, 2003b).

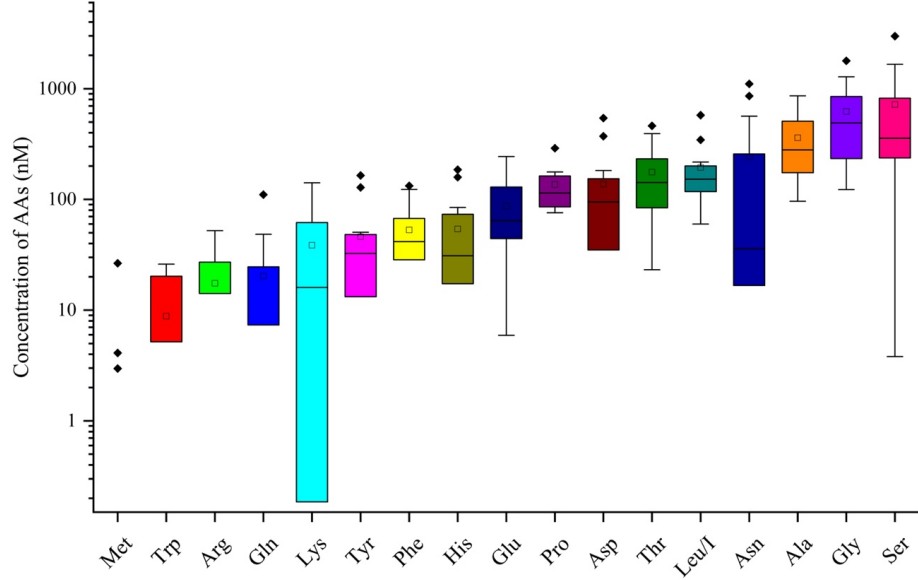


**Figure 2. Distribution of each AA for the 13 cloud samples. AA concentrations are in logarithmic scale. The bottom and top**
**lines of the box correspond to the 25[th] and 75[th] percentiles, respectively. The middle line represents the median value and the**
**square the mean value. The ends of the whiskers are the 10[th] and 90[th] percentiles, and the filled diamonds are outliers**
**(concentrations above the 90[th] percentile).**


**Figure 3. a. Distribution(nM) and b. relative contributions (% nM) of AAs molar concentrations in each cloud event sampled at PUY.**




Figure 3 illustrates for each cloud event the relative and absolute molar concentrations of AAs. As discussed above, the
TCAAs strongly vary between the different cloud events (Figure 3a). Their relative concentrations (Figure 3b) also vary
among the cloud samples. For example, Ser contribution exceeds 50 % in 25 Sep cloud, while Ser is almost absent in 11
Mar cloud sample, and vice versa for Ala. Asn prevails in 13 Jun and 24 Aug clouds. Nevertheless, the relative
concentrations are quite similar, and the highest TCAAs do not seem to be explained by the mere presence, in excess, of
a single AA.
Agglomerative hierarchical clustering (AHC) confirms these observations. AHC, used to categorize cloud samples based
on the AA concentration, successfully groups the 13 observations, with a satisfactory cophenetic correlation (correlation
coefficient between the dissimilarity and the Euclidean distance matrices) of 0.79 (Figure 4a).
The dotted line in Figure 4a represents the degree of truncation (dissimilarity = $5.7\ 10^6$) of the dendrogram used for
creating categories. This truncation is automatically chosen, based on the entropy level. The AHC profile plot (Figure 4b)
details the average composition of these two categories determined from the 18 AAs. In detail, the blue category gathers
10 cloud samples with lower AA concentrations. This blue category is the most homogeneous (within-class variance =
$3.7\ 10^5$), compared to the red category (within-class variance = $1.2\ 10^6$). Conversely, the red one, more heterogeneous,
gathers 3 cloud samples with higher AA concentrations except for Met (absent in most of the 13 samples). Note that the
13 Jun and 24 Aug cloud samples are isolated in the blue category due to their high Asn concentration.
These two AHC categories reflect the variability of some AAs in the 13 cloud samples. Because the computed p-value in
the Mann-Whitney test (Not shown) is lower than the significance level alpha = 0.05, the distribution of AA (Asp, Gly,
His, Leu/I, Lys, Phe, Ser, Thr and Tyr) concentrations can be accepted as significantly different between both AHC
categories.

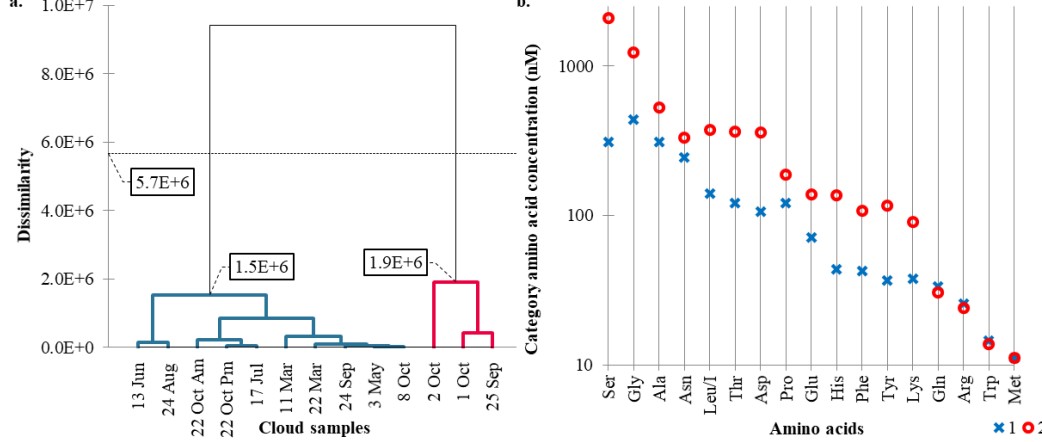


**Figure 4. a. Dendrogram representing the agglomerative hierarchical clustering (AHC) based on dissimilarities using the**
**Ward's method on concentrations of the 18 AAs. The 13 cloud samples are assigned to one of two established categories by**
**entropy (*i.e.*, dissimilarity < 5.7 E+6). b. Profile plot established by the AHC from the 18 main AAs. The Y axis, in logarithmic**
**scale, displays the average AA concentrations of the category.**

### 3.4 Comparison with previous studies on clouds, fogs, and rain

To our knowledge, only two studies refer to the AA characterization in cloud water (Table 2). A first one has been performed at PUY on 25 cloud samples; 16 AAs have been quantified by a different analytical procedure, using high-performance liquid chromatography connected to a fluorescence detection after derivatization of the AAs (Bianco et al., 2016b). They report a mean TCAA concentration of 2.67 µM with values ranging from 1.30 to 6.25 µM. These reported concentrations are within the same range of magnitude those of the present study (from 1.2 to 7.7 µM). However, the main difference between the present study and Bianco's study lies in the relative concentrations of the various AAs. Trp, Leu/I, Phe and Ser were the four most concentrated AAs (mean concentrations of 563, 548, 337 and 281 nM, respectively) while we found Ser, Gly Ala, and Asn as the most abundant AAs (mean concentrations of 721, 622, 360, and 264 nM, respectively). This discrepancy could result from the differences in the analytic tools, or on sampling characteristics, *i.e.*, cloud waters in the Bianco's study have been sampled during two short periods (March/April and November 2014), whereas in the present work, cloud waters have been collected over 6 years and covering different seasons.

The second one is a recent study reporting the characterization of AAs in cloud waters sampled at a marine site, the Cape Verde Islands (Triesh et al., 2021). Results also indicate variability of AA concentrations in cloud samples with values varying from 11.2 to 489.9 ng m$^{-3}$. These TCAAs are within the same range of magnitude as observed in this study (from 39 to 244 ng m$^{-3}$). In both studies (Cape Verde Islands and PUY), Ser, Ala and Gly are amongst the major AAs, but Asp is found to be highly concentrated in the Cape Verde's study. They also find that the relative distributions of these four AAs greatly change during the campaign period. Gly and Ser are found to be the dominant AAs in the first seven cloud samples, while Ala and Asp are also highly present together with Gly and Ser during the last part of the campaign (3 samples). They conclude that these differences are due to the different types of clouds sampled during this campaign. Triesch et al. (2021) show that some clouds present low TCAAs (less than 65 ng m$^{-3}$) with a dominance of Gly and Ser and a second group with elevated TCAAs (more than 250 ng m$^{-3}$) and Ser as major AA, followed by Ala and Gly. This enrichment of cloud waters in AAs could be due to oceanic sources or may be the result of *in situ* formation of AAs in cloud water by for example enzymatic degradation of proteins, as reported by the authors. These hypotheses are also supported by elevated concentrations of Asp at the end of the campaign that is a biologically produced AA. Globally, the concentrations and major groups of AAs reported by Triesch et al. (2021) agree with the present work. This can be explained by the remoteness of both locations and also the relevant marine influence encountered at PUY (Renard et al., 2020).

In fog waters, at Davis in Northern California, Zhang et al. (2003b) measured elevated concentration of TCAAs with a mean concentration of 20 µM. This is probably due to the proximity of the sampling site collection with local emission of aerosol particles in this rural environment; however, dominant AAs are the same (Ser, Gly, Ala, Asn and Leu/I). Two other studies in rainwater display similar AA concentrations and concentration ranking (Yan et al., 2015; Xu et al., 2019). The study in Korea measured lower AA concentrations (Free and combined AAs) at Seoul (an inland urban area) than those at Uljin (a coastal rural area) attributed to differences in contributing sources (Yan et al., 2015). A similar work has been performed at a suburban site in Guiyang (China) over one year and have shown a seasonal effect with a maximum level of AAs (Free and combined AAs) at spring and a minimal one at winter (Xue et al., 2019).

To conclude, few studies presented above report concentrations of AAs in cloud and fog waters. This is a challenging issue to compare those 3 studies that have been performed for contrasted environmental conditions and for a limited number of samples.





**Table 2. FAA concentrations in atmospheric aqueous samples: cloud, fog, and rain (n is relative to the number of the samples).**

| Localization | Environment/ medium | Period / Samples | Separation/ Detection Method | Concentrations of FAAs (Range and mean values) | Distribution Major FAAs | Reference |
|---|---|---|---|---|---|---|
| **Puy de Dôme Mountain, France (1465 m)** | Rural + marine influence (Cloud) | 03/2014 05-10/2018 09-10/2019 03-07/2020 13 samples | HPLC-MS/MS Standard addition | Range: 39 - 244 ng m$^{-3}$ | Ser > Gly > Ala > Asn > Leu/I | (This work) |
| **Puy de Dôme Mountain, France (1465 m)** | Rural + marine influence (Cloud) | 03-04/2014 (spring) 11/2014 (winter) 25 samples | HPLC-Fluorescence OPA-Derivatization | Mean: 118.6 ± 97.6 ng m$^{-3}$ | Trp > Leu/I > Phe > Ser | (Bianco et al., 2016b) |
| **Cabo Verde islands (744 m)** | Marine (Cloud) | 09-10/2017 (winter) 10 samples | HPLC-MS OPA-Derivatization | Range: 11.2 - 489.9 ng m$^{-3}$ | Ser > Asp > Ala > Gly > Thr | (Triesch et al., 2021) |
| **Northern California Davis, US** | Rural (Fog) | 1997 - 1999 (winter) 11 samples | HPLC-Fluorescence OPA-Derivatization | Mean: 40.8 ± 38.0 ng m$^{-3}$ (FAAs, protein type) | Ser > Gly > Leu > Ala > Val | (Zhang and Anastasio, 2003b) |
| **Atlantic Ocean, Golf Mexico (Cruise)** | Marine (Rain) | 09-10/1985 (n = 3) 02, 06, 09/1986 (n = 4) 7 samples | HPLC OPA/NAC-Derivatization | Mean: 604 ± 585 µg L$^{-1}$ | Gly > Ser > Ala > acidic AAs | (Mopper and Zika, 1987) |
| **Seoul, South Korea (17 m)** | Urban (Rain) | 03/2012 - 04/2014 36 samples | HPLC OPA-Derivatization | Mean: 21.0 ± 17.9 µg L$^{-1}$ | THAA: Gly > Glu, Ala, Asp, Ser | (Yan et al., 2015) |
| **Uljin, South Korea (30 m)** | Marine (Rain) | 02/2011 - 01/2012 31 samples | | Mean: 100.9 ± 110.2 µg L$^{-1}$ | | |
| **Guiyang, China (1300 m)** | Suburban (Rain) | 05/2017 – 04/2018 Summer (n = 29) Autumn (n = 9) Winter (n = 14) Spring (n = 13) 65 samples | HPLC OPA-Derivatization | Total range: 1.1 - 10.1 µM Mean: 3.7 µM Summer range: 1.3 - 6.6 µM Mean Summer: 2.9 µM Autumn range: 1.1 - 8.8 µM Mean Autumn: 4.4 µM Winter range: 1.5 - 9.9 µM Mean Winter: 3.4 µM Spring range: 2.6 - 10.1 µM Mean Spring: 5.2 µM | THAA: Glu + Gln, Gly, Pro > Asp, Ala | (Xu et al., 2019) |

• HPLC: High performance liquid chromatography
• OPA: Ortho-phthalaldehyde
• THAA: Total hydrolyzable amino acids
**3.5 Analysis of the environmental variability**
A large variability of the AA concentrations and relative proportions in the 13 cloud samples from PUY is observed
(Table S1). To better understand this variability, data are analyzed in parallel with various environmental factors such as
the air masses history and quantitative physical, chemical, and biological measurements. During their atmospheric
transports, the air masses received chemical species under various forms and from various sources, and could also undergo



multiphasic chemical transformations, as well as deposition. This section is devoted to the correlation between the AA
concentrations and the air mass history. To this end, PLS regressions are performed, and the results are validated with
nonparametric tests (Mann–Whitney tests).
The PLS matrix of the explanatory variables (the "Xs") is composed of the "zone matrix" (Sea/Continental surface
</> ABLH) from the CAT model, to which are added the temperature, the pH, the inorganic ion concentrations, the
bacteria density, the ATP concentration, and the seasons (Table S1). The matrix of the dependent variables (the "Ys") is
composed of the AA concentrations.
**Table 3. PLS correlation matrix between AA concentrations and "zone matrix" (Sea/Continental surface </> ABLH) from the**
**CAT model, and temperature, pH, cation and anion concentrations, TOC and $H_2O_2$ concentrations, bacteria density (CFU/mL)**
**and ATP concentration, and the seasons (Fall/Winter and Spring/Summer) determined from 13 cloud sampled at PUY. Highest**
**correlations are displayed in dark red and highest anti-correlations in dark blue.**

| Variables | Ser | Gly | Ala | Asn | Leu/I | Thr | Asp | Pro | Glu | His | Phe | Tyr | Lys | Gln | Arg | Trp | Met |
|---|---|---|---|---|---|---|---|---|---|---|---|---|---|---|---|---|---|
| Sea surface (< ABLH) | 0.88 | 0.68 | 0.18 | 0.38 | 0.76 | 0.82 | 0.74 | 0.31 | 0.53 | 0.70 | 0.84 | 0.71 | 0.58 | 0.21 | 0.20 | 0.08 | 0.08 |
| Sea surface (> ABLH) | -0.57 | -0.45 | 0.03 | -0.14 | -0.50 | -0.52 | -0.42 | -0.28 | -0.04 | -0.35 | -0.37 | -0.58 | -0.11 | 0.16 | 0.31 | 0.38 | 0.25 |
| Continental surface (< ABLH) | 0.54 | 0.33 | 0.45 | 0.26 | 0.57 | 0.63 | 0.55 | 0.63 | 0.33 | 0.56 | 0.22 | 0.68 | 0.18 | 0.17 | -0.48 | -0.06 | 0.00 |
| Continental surface (> ABLH) | -0.31 | -0.21 | -0.26 | -0.23 | -0.27 | -0.32 | -0.33 | -0.11 | -0.48 | -0.36 | -0.40 | -0.18 | -0.43 | -0.36 | -0.38 | -0.41 | -0.31 |
| Temperature (°C) | -0.08 | 0.29 | -0.14 | 0.29 | -0.10 | -0.15 | -0.07 | -0.25 | -0.41 | -0.28 | 0.40 | -0.13 | -0.36 | 0.51 | -0.13 | 0.09 | -0.20 |
| pH | -0.10 | 0.00 | -0.13 | -0.19 | 0.02 | -0.25 | -0.11 | -0.23 | 0.40 | 0.06 | -0.32 | -0.03 | 0.22 | 0.09 | 0.34 | 0.61 | 0.67 |
| $Na^+$ (µM) | 0.77 | 0.93 | 0.17 | -0.07 | 0.62 | 0.44 | 0.65 | 0.03 | 0.07 | 0.46 | 0.80 | 0.54 | 0.43 | 0.02 | 0.03 | 0.12 | -0.01 |
| $NH_4^+$ (µM) | -0.38 | -0.03 | -0.13 | -0.10 | -0.21 | -0.43 | -0.17 | -0.36 | -0.13 | -0.10 | -0.11 | -0.16 | -0.21 | 0.03 | -0.22 | -0.05 | 0.13 |
| $Mg_2^+$ (µM) | 0.06 | -0.11 | 0.09 | 0.10 | 0.17 | 0.11 | 0.15 | 0.23 | 0.46 | 0.41 | -0.02 | 0.31 | 0.18 | -0.11 | -0.08 | -0.08 | 0.30 |
| $K^+$ (µM) | 0.75 | 0.85 | 0.18 | 0.08 | 0.72 | 0.55 | 0.75 | 0.08 | 0.08 | 0.48 | 0.73 | 0.61 | 0.40 | 0.00 | -0.24 | -0.16 | -0.16 |
| $Ca^{2+}$ (µM) | -0.08 | -0.34 | 0.04 | 0.30 | -0.08 | 0.20 | 0.00 | 0.38 | -0.05 | 0.09 | -0.04 | 0.11 | -0.28 | -0.08 | -0.39 | -0.53 | -0.41 |
| $SO_4^{2-}$ (µM) | -0.11 | 0.12 | 0.37 | -0.29 | 0.13 | -0.11 | 0.12 | -0.01 | 0.41 | 0.18 | -0.15 | 0.08 | 0.13 | 0.10 | -0.13 | 0.21 | 0.58 |
| $NO_3^-$ (µM) | -0.17 | 0.00 | -0.05 | 0.02 | -0.01 | -0.03 | 0.02 | -0.12 | 0.02 | -0.07 | 0.01 | -0.05 | -0.21 | 0.07 | -0.26 | -0.39 | -0.05 |
| $Cl^-$ (µM) | 0.67 | 0.76 | 0.01 | -0.05 | 0.60 | 0.34 | 0.50 | 0.04 | 0.18 | 0.34 | 0.44 | 0.63 | 0.44 | 0.65 | 0.03 | 0.40 | 0.28 | 0.38 |
| TOC (mgC L$^{-1}$) | 0.03 | 0.09 | -0.09 | 0.25 | 0.04 | -0.10 | -0.10 | 0.15 | -0.35 | -0.09 | 0.16 | 0.19 | -0.26 | 0.04 | -0.27 | 0.06 | -0.16 |
| $H_2O_2$ (µM) | -0.19 | -0.01 | -0.44 | -0.10 | -0.26 | -0.46 | -0.40 | -0.32 | -0.26 | -0.25 | -0.24 | -0.15 | -0.16 | 0.00 | 0.21 | 0.43 | 0.20 |
| Bacteria (17°C; CFU/mL) | 0.28 | 0.07 | 0.50 | 0.20 | 0.41 | 0.51 | 0.41 | 0.33 | 0.43 | 0.44 | 0.21 | 0.36 | 0.36 | 0.11 | -0.19 | -0.39 | 0.19 |
| ATP (nM) | 0.13 | 0.58 | 0.00 | -0.14 | 0.19 | -0.04 | 0.22 | -0.32 | 0.05 | 0.19 | 0.33 | 0.17 | 0.01 | 0.19 | -0.16 | 0.14 | 0.16 |
| Fall / Winter | 0.44 | 0.40 | 0.46 | -0.44 | 0.32 | 0.36 | 0.36 | 0.29 | -0.15 | 0.15 | 0.21 | 0.25 | 0.21 | -0.39 | -0.23 | -0.31 | -0.25 |
| Spring / Summer | -0.44 | -0.40 | -0.46 | 0.44 | -0.32 | -0.36 | -0.36 | -0.29 | 0.15 | -0.15 | -0.21 | -0.25 | -0.21 | 0.39 | 0.23 | 0.31 | 0.25 |


The correlation matrix of this PLS (Table 3) displays significant (anti-) correlations. First, 9 of the 18 AAs (Gly, His, Tyr,
Asp, Leu/I, Thr, Phe and Ser) are robustly correlated with Sea surface under the atmospheric boundary layer height
(< ABLH), with correlation coefficients (r) ranging from 0.68 to 0.88. These 9 AAs are also significantly anticorrelated
with Sea surface in free atmosphere (> ABLH) (r ranging from -0.35 to -0.58), confirming direct influences from the
boundary layer. These 9 AAs coherently correlate with $Na^+$, $Cl^-$ and $K^+$ concentrations, confirming a marine influence for
those AAs, similar to the observations of Triesch et al. (2021).
To a lesser extent, the same tendency (correlation / anticorrelation) is observed with Continental surface
(< ABLH / > ABLH). PUY is a remoteness site and the presence of anthropic ions, such as $NO_3^-$ and $NH_4^+$, are correlated
with Continental surface (> ABLH) (Renard et al. 2020). Thereby, the AAs and, in particular, the 9 aforementioned AAs
are slightly anticorrelated with these anthropic ions.
No correlation appears between TOC concentration and the most abundant AAs, confirming the boundary layer influence,
as well as the variability of AA proportion in organic carbon. These 9 AAs are also slightly anti-correlated with $H_2O_2$
concentration, suggesting a potential influence of the photochemistry. The biological parameters, in particular the bacteria



density, are overall significantly correlated with the AA concentrations. The 9 AAs most correlated with the Sea surface
(< ABLH) are, to a lesser extent, also correlated with Fall/Winter.
To go further in modelling the environmental variability of the AA concentrations in our cloud samples, we performed a
simplified PLS restricting the Xs to the most correlated parameters (the PLS variable importance in the projection), *i.e.*,
the zone matrix. The index of the predictive quality of the models obtained with this PLS ($Q^2 = 0.19$ with one component)
is satisfactory given the complexity of the cloud composition. Figure 5 displays PLS correlation chart with t component
on axes t1 and t2. The main axis (t1) is linked to the ABLH and most of the AAs are correlated with < ABLH. The t2 axis
is linked to the zone (Sea / Continental surfaces) and it reveals a preponderance of marine influence, which is consistent
with the dominant western oceanic influence at PUY. Cloud water is a complex matrix resulting from the interaction of
many factors; cloud samples from more continental zones (northeast) could influence this model. Nevertheless, it appears
that the air mass history remains the prevailing parameter, after considering more cloud events, as observed in Renard et
al. (2020).

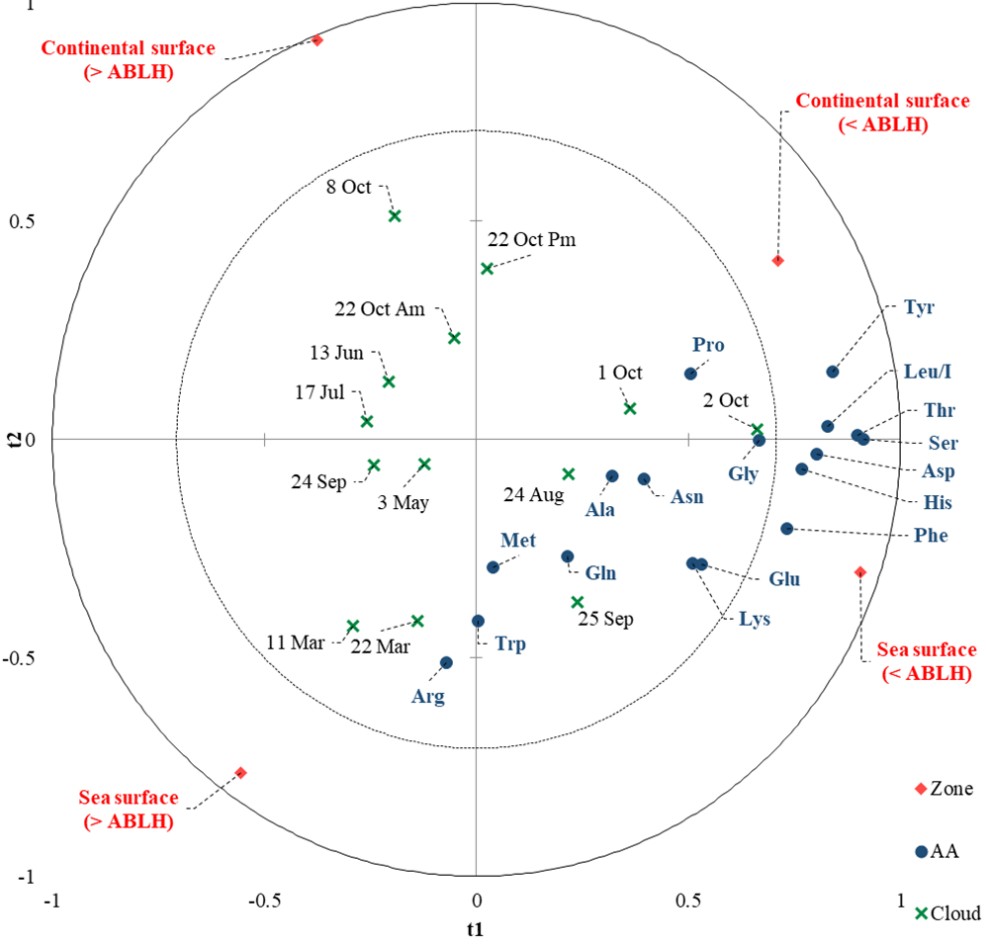


**Figure 5. Partial least squares (PLS) chart with t component on axes t1 and t2. The correlations map superimposes the**
**dependent variables from the chemical matrix (blue circles), the explanatory variables (red diamonds) and the cloud events**
**(green cross).**





The following section is devoted to the analysis of the processes occurring in the atmosphere that could potentially explain
the AA levels and distributions in the clouds sampled at PUY. These processes are linked to their sources and to their
potential biotic and/or abiotic transformations in the atmosphere.
**4.   Discussion**
Results reveal high variability in the relative concentrations of FAAs among cloud samples; however, some major FAAs
could be detected following this relative concentration ranking: Ser > Gly > Ala > Asn > Leu/I. On the contrary Trp and
Met present very low concentrations.
**4.1  Potential influence of the initial AAs distribution in biological matrices**
As free AAs are mostly from biological origin, we first compared the AA composition of various biological
macromolecules (proteins, peptidoglycans, *etc*.) that can be the source of AAs after hydrolysis with the relative
concentrations of AAs measured in the studied cloud samples.
A study reports the relative distributions of AAs in proteins extracted from bacterial cells (*S. aureus* and *E.coli*) and from
various mammalian cells (Okayasu et al., 1997). Although there are some differences between mammalian and bacterial
protein composition, some AAs are clearly dominant. As a first approximation, we can share AAs in two groups: Ala,
Gly, Asp+Asn, Glu+Gln, Val, Leu/I, Lys are more abundant compared to others (Ser, His, Arg, Pro, Tyr, Met, Cys, Phe).
Globally, this relative abundance of AAs initially constituting proteins presents similarities with the relative
concentrations present in our samples. In particular, Ala, Gly, Leu/I and Asn/Asp are the most abundant in our samples
as in the proteins. It is also true for Met which abundance is low. However, Ser proportion is clearly less dominant in
proteins in comparison with the proportion encountered in the studied cloud samples. Ser is also a dominant AA in other
atmospheric samples (Table 3), revealing another sources or processes.
We looked at the composition of peptidoglycan that form the cell wall of microorganisms that can be an atmospheric
source of AAs. Peptidoglycan monomers consist in two joint amino sugars (Nag, Nam) connected with a pentapeptide
containing the L-Ala, D-Gly L-Lys, D-Ala, D-Ala sequence. Peptidoglycans can thus represent a major source of Ala and
Gly, that are the major AAs detected in our samples.
Finally, we searched for the potential origin of Ser. Hecky et al. (1973) reports the AA composition of cell walls from six
different diatom species, selected on the basis of taxonomy and habitat diversity. Three are from estuarine origin, the
other from fresh waters. The protein template of these cell walls is composed of the following AA sequences: Asp-Ser-
Ser-Gly-Thr-Ser-Ser-Asp-Ser-Gly. Ser is thus highly abundant in these aquatic organisms and plays an important role in
the complexation of the silicon ($Si^{4+}$). AAs in sea water during phytoplankton blooms were also investigated (Ittekkot,
1982): Glu concentration is maximum in the early stages of the bloom, while Asp, Gly, Ala and Lys concentrations
increase at the end of the bloom. In parallel, Ser was one of the most abundant AA and its concentration remains high all
along the bloom period. Ser could come from the cell walls of some phytoplankton species which are diatoms. Hashioka
et al. showed that diatoms could contribute up to 80% of the total phytoplankton in the ocean to during bloom events
(Hashioka et al., 2013). The high Ser concentration measured in our cloud samples could thus originate from diatoms and
could be a marker of their oceanic origin.



In the following, we aim at discussing more the variability of the AAs distributions and concentrations among the samples
looking at the air mass history (*i.e.*, sources) and their atmospheric transformations.
**4.2   Potential influence of the air mass origin on the AA concentrations and their relative distribution**
Table S4 summarizes the studies that analyze AAs quantity and distribution in various atmospheric media. Interestingly,
the systematic presence of Ser, Ala and Gly is observed in the various atmospheric waters including clouds (Triesch et
al., 2021), fogs (Zhang and Anastasio, 2003b) and rains (Mopper and Zika, 1987; Yan et al., 2015). These 3 AAs are also
significantly present in aerosols over contrasted regions over the world: rural sites (Zhang and Anastasio, 2003b) marine
sites (Matsumoto and Uematsu, 2005; Triesch et al., 2021; Violaki and Mihalopoulos, 2010; Wedyan and Preston, 2008),
urban or suburban sites (Barbaro et al., 2011; Samy et al., 2013) and polar sites (Scalabrin et al., 2012).
Looking more specifically at only two AAs (Gly and Ala), this list of studies can be extended to other works: in rain (Xu
et al., 2019), in marine aerosols (Mace et al., 2003b; Mandalakis et al., 2011), in rural aerosols (Ruiz-Jimenez et al., 2021;
Samy et al., 2011), in polar and remote sites (Barbaro et al., 2020; Barbaro et al., 2015; Feltracco et al., 2019). We can
notice that Gly is globally one of the major FAAs in all the reported studies (see Table S4 and joint explanations). In the
present study, we detect significant concentrations of Leu/I in agreement with only 3 other studies (Bianco et al., 2016b;
Mashayekhy Rad et al., 2019; Wedyan and Preston, 2008).
We overall found the same major groups of AAs that are commonly detected in marine clouds and aerosols. However,
one of the main differences is the high concentration of Asn in two of our samples, instead of the more common Asp,
suggesting potential conversion Asp/Asn (Jaber et al., 2021), and indicating that the origin of the clouds and aerosols is
not the only main driving factor explaining the final observed FAAs relative proportion in the clouds sampled at PUY.
Moreover, the presence of similar trends of AA composition in aerosols sampled under different sites (rural, marine,
urban, and polar) and in our cloud samples shows various influences from both continental and marine sources.
In agreement with the results of the PLS analysis (Figure 5), a significant correlation ($r = 0.78$) is observed between the
TCAA and the time spent by the air mass over the sea and under the boundary layer (Sea surface < ABLH). The correlation
between the TCAA and Sea and the Continental surfaces (< ABLH) is even higher ($r = 0.86$) (Figure S5), indicating that
the boundary layer influences the total amount of AAs rather than their relative concentration. When the air mass is
transported in the free troposphere, the TCAA is lower, possibly because of the remoteness of the direct sources and
because of chemical transformations that might be more intense in this upper part of the atmosphere.
To go further, Triesch et al. (2021) compared the AA composition of samples collected at Cabo Verde (marine
environment) in both aerosol and cloud phases. They show that FAAs are partitioned according to their hydropathic
properties. They show that the hydrophobic AAs (Ala, Val, Phe, Leu/I) represent a much lower proportion (about 25 %)
of the total AAs present in cloud water, compared to the neutral (Ser, Gly, Thr, Pro, Tyr) plus the hydrophilic AAs (Glu,
Asp, Gaba). Figure 6 shows the distribution of the AAs in our samples collected at the PUY station according to their
hydrophobic *versus* hydrophobic + neutral properties. Clearly the concentrations of hydrophilic (Glu, Asp, Gln, Asn, His,
Lys, Arg) and neutral (Trp, Tyr, Gly, Thr, Ser, Pro) AAs are much higher (average value of 80 %) than that of hydrophobic
(Leu/I, Phe, Met, Ala) ones in all the samples, except in the 11 Mar sample where the hydrophilic + neutral AAs represent
only 40.8 % of the total FAAs. Our results are consistent with those measured in cloud samples at Cabo Verde; this
suggests that the hydrophobic nature of AAs is less favorable for their incorporation in cloud droplets due to their low
solubility.

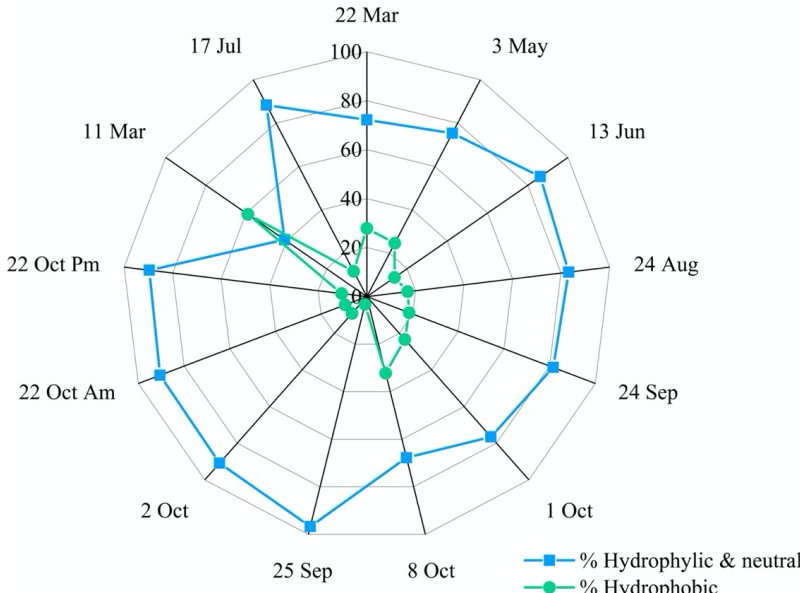


**Figure 6. Relative composition of AAs grouped by hydropathy (hydrophilic + neutral versus hydrophobic AAs) observed in each cloud sample.**

Although the initial AA composition of the emitted aerosols can greatly impact the type of FAAs, the aging of the samples
due to biotic and abiotic processes must be considered to explain the presence of major or minor groups of AAs.

## 4.3 Potential influence of the atmospheric aging of AAs

Table 4 reports calculated, and experimental lifetimes of the different AAs targeted in this work considering different
biotic and abiotic processes.
First, AA theoretical lifetimes are calculated considering the reactivity constants of AAs with HO$^\bullet$ radicals, O$_3$ and $^1$O$_2^*$.
The values issued from the work of Jaber et al. (2021), Triesch et al. (2021) and Mc Gregor and Anastasio (2001) are
reported in columns A, B and C, respectively. At first glance it can be noticed that the lifetimes depend on the AAs and
can vary from a few hours or even minutes to a few days. Globally, reported values from the three studies are rather
consistent although they were calculated using a different set of reactivity constants and different oxidant concentrations
(see foot notes of Table 4). These AA theoretical lifetimes could explain the very low Met and Trp concentrations
measured in our cloud samples which are very reactive, and the large one measured for Gly and Asn and in some extent
for Ser and Ala which are very slowly transformed. However, they do not fit with the large amounts of Leu/I, except for
the values given by Mc Gregor and Anastasio (2001).





**Table 4. Estimated atmospheric lifetimes of AAs degraded by atmospheric biological and chemical processes. AAs are classified**
**following their mean concentrations measured in the present study. A brief description of the calculations is added below this**
**table. More information can be found in SI for the calculations performed in this study based on the work from Jaber et al.**
**(2021).**

| AAs | Theo. lifetimes by oxidation (days) | Theo. lifetimes by oxidation (days) | Theo. lifetimes by oxidation (days) | Exp. lifetimes by oxidation processes (days) | Exp. lifetimes by oxidation processes (days) | Exp. lifetimes by biological processes (days) |
|---|---|---|---|---|---|---|
| **Reference** | This study (adapted from Jaber et al. (2021)) | Triesch et al. (2021) | Mc Gregor and Anastasio (2001) | This study (adapted from Jaber et al. (2021)) | Mc Gregor and Anastasio (2001) | This study (adapted from Jaber et al. (2021)) |
| **Degradation processes** | Oxidants: HO$^\bullet$, O$_3$ and $^1$O$_2^*$ | Oxidant: HO$^\bullet$ | Oxidants: HO$^\bullet$, O$_3$ and $^1$O$_2^*$ | Irradiation experiments in artificial cloud medium | Irradiation experiments in fog waters | 4 microbial strains in artificial cloud medium |
| **Additional information** | (A) | (B) | (C) | (D) | (E) | (F) |
| **Ser** | 4.47 | 1.64 | / | 17.55 | > 3.67 | 0.63 (15.1h) |
| **Gly** | 41.26 | 3.09 | > 170 | $ | > 3.67 | 4.20 |
| **Ala** | 4.16 | 6.83 | / | 22.60 | > 3.67 | 0.31 (7.6h) |
| **Asn** | 22.05 | / | / | $ | > 3.67 | 0.34 (8.1h) |
| **Leu/I** | 0.64 (15.4h) | 0.29 | 6.67 | 43.34 | 4.2 | 7.09 |
| **Thr** | 2.21 | 1.03 | / | 4.67 | / | 1.28 |
| **Asp** | 22.47 | / | / | $ | 2.42 | 1.55 |
| **Pro** | 1.72 | 1.70 | / | $ | / | 0.31 (7.4h) |
| **Glu** | 5.49 | 3.29 | 37.5h | 17.64 | 2.25 | 0.19 (4.5h) |
| **His** | 0.10 (2.5h) | / | 0.2 (5h) | 22.60 | 1.00-1.83 | 1.79 |
| **Phe** | 0.17 (4.2h) | 0.08 | 1.75 | / | > 3.67 | 1.80 |
| **Tyr** | 0.08 (2.0h) | 0.04 | 0.05 (1.2h) | 3.56 | 1.25-2.33 | 0.86 (20.5h) |
| **Lys** | 3.25 | / | / | $ | > 3.67 | 0.59 (14.3h) |
| **Gln** | 2.13 | 0.97 | / | $ | / | 0.20 (4.8h) |
| **Arg** | 0.32 (7.7h) | / | / | $ | > 3.67 | 0.37 (9h) |
| **Trp** | 0.06 (1.4h) | / | 0.01 (0.15h) | 10.51 | 0.11-0.38 | 6.97 |
| **Met** | 0.01 (0.13h) | 0.06 | 0.01 (0.24h) | 6.23 | 0.07-0.52 | 2.75 |

(A): Theoretical calculations considering kinetic rate constants for the AAs oxidation by HO$^\bullet$. O$_3$ and $^1$O$_2^*$ following Jaber et al. (2021). Aqueous
concentrations of HO$^\bullet$, O$_3$ and $^1$O$_2^*$ are respectively equal to $10^{-14}$, $5.0 \times 10^{-10}$ and $1.0 \times 10^{-12}$ M.
(B) Theoretical calculations by Triesch et al. (2021). The mean lifetimes are estimated by considering pH-dependent rate constant of AAs with HO$^\bullet$.
An HO$^\bullet$ concentration of $2.2 \times 10^{-14}$ M is considered in this study.
(C) Theoretical calculations by Mc Gregor and Anastasio (2001) were done under typical midday, wintertime conditions. Several oxidants were
considered: the photoproduction of HO$^\bullet$ and $^1$O$_2^*$ in the droplets, the source of HO$^\bullet$ and O$_3$ in the aqueous phase by mass transfer.
(D) Experimental irradiation of 19 AAs at a concentration of 1 µM each in an artificial cloud medium were conducted. HO$^\bullet$ production was performed
using Fe-Ethylenediamine-N,N'-disuccinic acid (EDDS) complex solution. HO$^\bullet$ concentration of $8.3 \times 10^{-13}$ M is estimated. $: Lifetimes cannot be
calculated since a production is observed during the experiment.
(E) Irradiation experiments using simulated sunlight illumination were performed on real fog waters containing AAs.
(F) Biodegradation experiments of 19 AAs were performed by Jaber et al. (2021) using 4 microbial strains (*Rhodococcus enclensis* PDD-23b-28,
*Pseudomonas graminis* PDD-13b-3, *Pseudomonas syringae* PDD-32b-74 and *Sphingomonas sp.* PDD-32b-11) in artificial cloud water.

A second approach is to consider transformation rate measurements to further calculate experimental lifetimes.
Experimental investigations were designed to evaluate both abiotic and biotic processes. Photodegradation experiments
have been designed to assess oxidations processes, the first one was performed by Jaber et al (2021) in a microcosm
mimicking cloud environment using artificial cloud medium (Table 4, Column D), the second one (Mc Gregor and



Anastasio, 2001) consisted in irradiating real fog samples (Table 4, column E). In both cases the HO• concentration is
quantified. Interestingly the obtained experimental lifetimes are globally longer than the theoretical ones, many of them
exceeded 3 days and only Trp and Met lifetimes in the fog experiment are less than one hour. Moreover, certain AA
lifetimes could not be calculated from transformation rates measured by Jaber et al. (2021) as they observed production
and not degradation of some AAs (Gly, Asn, Asp, Pro, Phe, Lys, Arg, Gln). These experimental results, which are different
from theoretical ones, reflect a much higher complexity of the occurring transformations. On the one hand, irradiations
are performed on complex media containing a mixture of AAs, as well as other carbon and nitrogen sources. So, the
measured transformation rates are net values reflecting both synthesis and degradation processes, and even potential inter-
conversion mechanisms. On the other hand, theoretical lifetimes are calculated from reactivity constants measured in pure
water containing a single AA without any other C or N components and thus far from the chemical reactivity in real
environmental samples. In addition, it is difficult to interpret this data in more detail. Indeed, very few studies have studied
the photo-produced compounds during these oxidation processes. Some works report the formation of carboxylic acids,
nitrate and ammonia from AAs photo-transformations, or the conversion of AAs in another different AAs (His to Pro,
Asp and Asn, Phe to Tyr, Pro to Glu) (see Jaber et al. (2021) for review). More detailed pathways of abiotic
transformations are only available for Trp, Tyr and Phe (Bianco et al 2016a; Pattison et al., 2012). In spite of this complex
situation, the long lifetimes or net production of Ser, Leu/I, Gly and Asn (Table 4, columns D and E) could explain the
relative high concentrations of these compounds in cloud waters collected at PUY. On the contrary the short lifetimes
(< 1 day) measured in fogs could explain the low concentrations of Met and Trp. However, the lifetimes reported here
cannot fully explain intermediate concentration values measured for most of the other AAs; more work is needed to better
understand oxidation pathways in complex atmospheric media and measure additional transformation rates.
Potential biological transformation processes have been also evaluated in the lab. Recent work suggests that
biodegradation of AAs could occur in rain and in aerosols; these hypotheses are based on Degradation Index (DI)
calculations (Xu et al., 2020; Zhu et al., 2021). To calculate biotransformation lifetimes, transformation rates were
measured in microcosms with 4 bacterial strains isolated from clouds and representative of this medium and incubated in
artificial cloud water (Jaber et al., 2021). As in the previous case of irradiation in the same microcosm, it was shown that
some AAs could be degraded but also produced depending on the bacterial strain. The resulting biodegradation rates were
thus calculated considering the proportion of each type of cell in real cloud (see Table S4 for more details). From these
global reaction rates, lifetimes could be calculated for individual AA (Table 4, Column F). These biological lifetimes are
very different from those obtained considering oxidation processes, and globally much shorter. *Per se* they cannot explain
the ranking of the larger AAs (Ser, Ala, Leu/I, Asn) and lower AAs (Met, Trp) concentrations in our cloud samples,
suggesting they might not be the major contribution to the transformation of these AAs. However, when other compounds
are considered with rather low concentrations such as Gln, Arg, Lys, Phe, His, experimental oxidation lifetimes are long
while biodegradation lifetimes are much shorter, combination of these two processes could reflect a more realistic
situation. Biosynthesis and biodegradation pathways are very complex and interconnected and are fully described in
databases (see: https://www.genome.jp/kegg/pathway.html). The complexity comes from the behavior of the different
microorganisms to use these pathways. Up to now the only biodegradation rates related to atmospheric waters are from
Jaber et al. (2021) and might be incomplete; more experimental work should be conducted on real atmospheric samples.



**Conclusion**

This study reports the quantification of amino acids in cloud waters sampled at the puy de Dôme station using a new approach based on a direct *in situ* analysis of the sample. Concentration of AAs represent in average nearly 2 % of the TOC with a significant variability of TCAAs among the different samples. This heterogeneity is also observed in the AAs distribution between the samples, but certain AAs are more dominant, especially Ser, Gly, Ala, Asp and Leu/I. These AA relative proportions can be explained by the original biological matrices that emit AAs into the atmosphere, by the hydrophilicity of AAs that favours their incorporation in the cloud water and finally by their transformations during their transport into the atmosphere that modulate the total of AAs. At PUY, the residence of the air masses within the boundary layer, especially above the sea, seems also to surely increase the total amount of AAs in cloud water. Conversely, the AA concentrations seem to decrease when the photolysis conditions are more favourable (free troposphere or Spring / Summer period). In other words, the AAs concentration is modulated by the sources (mainly from the boundary layer) and the sinks associated with the photodegradation.

However, it is still hard to validate all the formulated hypotheses that have been proposed to explain the differences in the amounts and proportions of the various AAs found in our samples. This variability integrates many factors that are interconnected or decorrelated and that should be investigated in the future. Some future targeted works could be mentioned. First, this study is to our knowledge only the third one performed on cloud samples. More samples should be collected at different seasons and at other sites presenting contrasted environmental conditions. This is crucial to robustly evaluate atmospheric AA variability considering the effect of difference sources and atmospheric transport. Second, a major limitation encountered to interpret the impact of transformation processes on the final distribution of AAs in atmospheric samples lies on the lack of knowledge available in this field. Very few studies report the complex mechanisms of biotic and abiotic transformations of AAs under realistic atmospheric conditions. Photochemists and biologists should develop interdisciplinary work to describe these transformation pathways; this remains a challenging task.

**Author contributions.**

A.-M. D. & L. D. designed the project. A. B., M. B. & L. D. sampled the clouds at PUY. M. B., S. J. and M. L. conducted the analysis. J.-L. B. used the CAT model to calculate backward trajectories and the "matrix zone". P. R. and F. R. performed the statistical analysis. P. R., F. R., A.-M. D. & L. D. wrote the paper.

**Competing interests.**

The authors declare that they have no conflict of interest.

**Acknowledgements.**

This work was funded by the French National Research Agency (ANR) in the framework of the 'Investment for the Future' program, ANR-17-MPGA-0013. S. Jaber is recipient of a grant from the Walid Joumblatt Foundation for University Studies (WJF), Beirut, Lebanon, and M. Brissy from Clermont Auvergne Metropole. CO-PDD is an instrumented site of the OPGC observatory and LaMP laboratory, supported by the Université Clermont Auvergne (UCA), by the Centre National de la Recherche Scientifique (CNRS-INSU), and by the Centre National d'Etudes Spatiales (CNES). The authors are also very grateful for the financial support from the Fédération des Recherches en





591 Environnement through the CPER funded by Region Auvergne – Rhône-Alpes, the French ministry, ACTRIS Research

592 Infrastructure, and FEDER European Regional funds. The authors also thank the I-Site CAP 20-25.




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
