# Peer review of "Free amino acids quantification in cloud water at the puy de Dôme station (France)"

_Atmospheric Chemistry and Physics, 2021_

## Author Comment (AC1)

**Responses to the reviewer 1 comments**

Renard et al. present interesting information about the composition of free amino acids (AA) in cloud water from a French mountain site. The LC-MS analytical work is sophisticated and sensitive. Care is taken to avoid important artifacts associated with matrix effects commonly found in LC-MS analysis of complex environmental samples. The results are novel, and the authors do a good job exploring several hypotheses related to factors influencing the abundance and relative composition of the observed AA. I do have several suggestions to improve the manuscript prior to publication.

We would like to thank the reviewer for this positive comment and for the suggestions. This will improve the quality of the manuscript. We answer to your comments below.

1. Please provide more detail about experimental methods:

- Add a short description of the cloud water collectors

We modified the text as follows:

Line 115:

*"The sampling is performed using aluminum cloud water collectors under non-precipitating and non-freezing conditions as described in Deguillaume et al. (2014). Cloud droplets are collected by impaction onto a rectangular plate which then flows directly into a sterilized bottle going through a funnel. The impactor has an estimated cut-off diameter of 7 μm. Before cloud collection, cloud impactors are cleaned using milliQ water and sterilized by autoclaving. Immediately after sampling, a fraction of the aqueous volume is filtered using a 0.2 μm nylon filter (Fisherbrand™) to eliminate microorganisms. The samples are then stored in the dark and frozen at -20 °C (ATP, ion chromatography, total organic carbon, and amino acids). For cell counts, samples are stored at 4 °C after adding a fixative. The analyzes are performed shortly thereafter."*

- Provide more detail about the instruments and materials used, such as the manufacturer of the nylon filter and the model of the Shimadzu TOC analyzer

We modified the manuscript as follows:

Line 119:

*"0.2 μm nylon filter (Fisherbrand™)"*

Line 128:

*"TOC analyses are performed with a TOC analyzer (Shimadzu, TOC-5050A)."*

- Line 118: be more specific about storage temperature and storage times prior to analysis for particular analytes, especially the AAs.

For the measurements of ATP and chemical compounds (ionic chromatography, total organic carbon, and amino acids), samples were kept in the dark at -20 °C before their analyses in the following months. For the counting of bacterial cells, samples were stored in the dark at 4 °C after the addition of a fixative agent (glutaraldehyde, 0.5 % final) before their analyses in the following weeks.

We modified as follows:

Line 120:

*"The samples are then stored in the dark and frozen at -20 °C (ATP, ion chromatography, total organic carbon, and amino acids). For cell counts, samples are stored at 4 °C after adding a fixative. The analyzes are performed shortly thereafter."*

- Please add a reference for the Gerber Scientific PVM-100.

The reference *Gerber* (1991) has been added in the manuscript.

Line 133:

*"Gerber, H.: Direct measurement of suspended particulate volume concentration and far-infrared extinction coefficient with a laser-diffraction instrument, Appl. Opt., 30, 4824-4831, 1991."*

2. The authors compare AA compositions of cloud water samples collected over 6 years. Are they confident that differences in AA abundance and relative composition are not affected by differing storage times/conditions?

Cloud samples are filtered to remove microorganisms before dark storage at -20 °C. Under these conditions, biotic or abiotic transformations should be negligible. In addition, as mentioned in line 120, the time between cloud collection and their analyzes was constant throughout the sampling period. With the exception of the 22-Mar-14 sample, AA measurements were conducted a few months after sampling. We kept the latter because it did not show significantly lower AA concentrations (Kruskal-Wallis test, p-values <0.05) than the samples from 2018, 2019 and 2020.

3. Line 223 and abstract: Standard addition prevents introduction of analytical biases resulting from matrix effects. It does not actually prevent the matrix effects.

*You are right, the term "avoid" is probably excessive. Hewavitharana et al.* (2018) used the term "overcome".

We modified as follows:

Line 14:

*"This quantification has been performed without concentration neither derivatization, using LC-MS and the standard addition method to correct for matrix effects."*

Line 101:

*"In addition, to overcome matrix effect, we propose to quantify the AAs by the standard addition method (Hewavitharana et al., 2018)."*

Line 230:

*"The standard addition method also restrains matrix effects which are very commonly encountered with environmental matrices (Hewavitharana et al., 2018)."*

4. The authors need to better distinguish the two sections of the manuscripts discussing STD. On p. 7, it would help the reader if they clearly indicated that they are speaking about the precision of the measurements of AA while they later discuss the variability of concentrations across different cloud samples. In both places the authors tend to rely on the jargon of discussing STD. More nuance in the descriptions would help.

To distinguish the standard deviation describing the precision of the measurements ($STD_M$), from the standard deviation describing the variability of concentrations across different cloud samples (STD), we replaced STD by $STD_M$ in the section 3.1 ("Evaluation of LC-MS technique for a direct measurement of AAs in cloud"), and modified the text as follows:

Line 232:

*"Concentrations values obtained for all AAs and cloud samples – as well as the standard deviation of the measurements ($STD_M$) (i.e., the precision of the measurements of AA concentrations) – are reported in Table S3 and detailed in Figure S3."*

5. The authors are correct that not many publications have reported concentrations of speciated AA in cloud, fog, or rain. They did, however, overlook an early, seminal paper by Mopper and Zika (1987) Nature 325, 246-249. They should review this early paper, add it to their comparison tables, and include its findings in their discussions of comparisons to their current work.

*You are right. Mopper and Zika (1987)* reported AA concentrations in their paper. We had taken their results into account, and we had mentioned them in the introduction (lines 70 and 95), in the section 3.4 ("Comparison with previous studies on clouds, fogs, and rain") (Table 2), in the section 4.2 ("Potential influence of the air mass origin on the AA concentrations and their relative distribution") (line 465) and in the SI (Table S4).

6. Please replace "hydropathy" with hygroscopicity throughout. Replace "multiphasic" with multiphase.

This has been replaced in all the manuscript.

7. The authors switch between referring to the Cape Verde Islands and the Cabo Verde Islands. Please switch all to Cape Verde Islands.

We have replaced "Cabo Verde" by "Cape Verde".

8. Please change terminology so that you refer to trajectories within (not below) the boundary layer.

We used either "below the ABLH" (consistent with the model), or "inside the ABL".

9. Liquid chromatography is coupled to mass spectrometry, not "hyphened to" it.

We modified the text as indicated.

10. The manuscript contains numerous errors in grammar and syntax and several awkwardly phrased sentences. There are also unusual choices to capitalize certain words (e.g., Sea, Free, Continental, all of which should not be capitalized), some misspellings (e.g., De Hann), many cases of singular-plural disagreements between nouns

and verbs, and numerous poor choices of prepositions. With a few exceptions, the authors' meaning is clear, but the text would greatly benefit from English language editing.

We will contact after the reviewing process the ACP editing service to improve the quality of the manuscript. Therefore, the errors in grammar and syntax will be corrected.

**References**

Deguillaume, L., Charbouillot, T., Joly, M., Vaïtilingom, M., Parazols, M., Marinoni, A., Amato, P., Delort, A.-M., Vinatier, V., Flossmann, A., Chaumerliac, N., Pichon, J. M., Houdier, S., Laj, P., Sellegri, K., Colomb, A., Brigante, M., and Mailhot, G.: Classification of clouds sampled at the puy de Dôme (France) based on 10 yr of monitoring of their physicochemical properties, Atmos. Chem. Phys., 14, 1485–1506, https://doi.org/10.5194/acp-14-1485-2014, 2014.

Gerber, H.: Direct measurement of suspended particulate volume concentration and far-infrared extinction coefficient with a laser-diffraction instrument, Appl. Opt., AO, 30, 4824–4831, https://doi.org/10.1364/AO.30.004824, 1991.

Hewavitharana, A. K., Abu Kassim, N. S., and Shaw, P. N.: Standard addition with internal standardisation as an alternative to using stable isotope labelled internal standards to correct for matrix effects-Comparison and validation using liquid chromatography-tandem mass spectrometric assay of vitamin D, J Chromatogr A, 1553, 101–107, https://doi.org/10.1016/j.chroma.2018.04.026, 2018.

Mopper, K. and Zika, R. G.: Free amino acids in marine rains: evidence for oxidation and potential role in nitrogen cycling, 325, 246–249, https://doi.org/10.1038/325246a0, 1987.

---

## Author Comment (AC2)

**Responses to the reviewer 2 comments**

Renard et al. present a very interesting study about free amino acids in cloud water at puy de Dome. Such data are rare and strongly needed. The data are well-presented and discussed. The general style is good and the English very well readable.

However, several things should be considered before publication:

*We would like to thank the reviewer for the proposed corrections. This will improve the quality of the manuscript. We answer to your comments below.*

- Line 105 and following: The samples are spread over a wide temporal range. First of all, please add the year, whenever you are referring to a sample in the Tables and in the text (not just day and month), as several years are regarded.

*You are right. We added the years to the names of the samples. Therefore, we modified the labels in the text, Figures and in the SI.*

- What about possible artefacts, especially due to the long sample storage time? Could you add more details how the samples were treated (*e.g.*, filtered before freezing)?

*For the measurements of ATP and chemical compounds (ionic chromatography, total organic carbon, and amino acids), samples were kept in the dark at -20 °C before their analyses in the following months. For the counting of bacterial cells, samples were stored in the dark at 4 °C after the addition of a fixative agent (glutaraldehyde, 0.5 % final) before their analyses in the following weeks.*

*We modified as follows:*

*Line 120:*

*"The samples are then stored in the dark and frozen at -20 °C (ATP, ion chromatography, total organic carbon, and amino acids). For cell counts, samples are stored at 4 °C after adding a fixative. The analyzes are performed shortly thereafter."*

- Did this sample treatment or storage affect e.g. biological measurements? Did you check this?

*Cloud samples are filtered to remove microorganisms before dark storage at -20 ° C. Under these conditions, biotic or abiotic transformations should be negligible. In addition, as mentioned in line 120, the time between cloud collection and their analyzes was constant throughout the sampling period. With the exception of the 22-Mar-14 sample, AA measurements were conducted a few months after sampling. We kept the latter because it did not show significantly lower AA concentrations (Kruskal-Wallis test, p-values < 0.05) than the samples from 2018, 2019 and 2020.*

- Line 114: Please add information on the collection efficiency of the cloud water sampler.

*We modified the text as follows:*

*Line 115:*

*"The sampling is performed using aluminum cloud water collectors under non-precipitating and non-freezing conditions as described in Deguillaume et al. (2014). Cloud droplets are collected by impaction onto a rectangular plate which then flows directly into a sterilized bottle going through a funnel. The impactor has an estimated cut-off diameter of 7 μm. Before cloud collection, cloud impactors are cleaned using milliQ water and sterilized by autoclaving. Immediately after sampling, a fraction of the aqueous volume is filtered using a 0.2 μm nylon filter (Fisherbrand™) to eliminate microorganisms."*

- Line 183/184: You report the linearity ($R^2$), However, this value alone is not sufficient to describe the correlation, as e.g., a good linearity can be obtained by a good fit of just one high calibration point. Please consider to add values that describe the significance (such as the p-value).

We absolutely agree on this point. The p-values are essential to describe the significance of these calibrations. We did not add them for clarity, but the p-values of all the correlations are strictly less than 0.05. We modified the text as follows:

*"Table S3. Concentration (μg L$^{-1}$ with dilution 9:1, detailed in Figure S3), calibration curve and R² data for the 18 amino acids (AA) analyzed in the 13 clouds sampled at PUY. The calculation method (detailed in Figure S3) might mathematically provide negative values for the concentration. Approximating the limit of quantification by the standard deviation (STD), amino acids with an average concentration lower than STD are considered unquantifiable. ND: Not Determined. The p-values of all the correlations are strictly less than 0.05."*

- You show only one calibration curve of the standard addition (Fig. S3) of a sample containing high concentrations. I am wondering how it looks like for lower concentrated samples.

With $R^2 > 0.99$ and p-values $\ll 0.05$, all calibration curves have the same statistical significance. However, the graphic representation is less didactic for lower concentrated samples. For example, comparing Gly and Asn concentrations in the 11-Mar-20 cloud sample (Figures below), we observe that the X-intercept is less separate for the less concentrated AA (Asn).

[Figure]

*Asn Calibration curve (11-Mar-20 cloud sample)*

[Figure]

*Gly Calibration curve (11-Mar-20 cloud sample)*

- Line 200: Why do "marine" clouds have a low ion concentration? I´d expect some levels of sea salt.

The "Marine" term could be confusing.

As detailed in Renard et al. (2020), we performed agglomerative hierarchical clustering (AHC), an iterative classification method, on 300 cloud events. The aim was to define homogeneous categories based solely on their chemical concentrations ($Cl^-$, $Mg^{2+}$, $Na^+$, $NH_4^+$, $NO_3^-$ and $SO_4^{2-}$).

The category with low ion concentrations ($Cl^-$, $Mg^{2+}$, $Na^+$, $NH_4^+$, $NO_3^-$ and $SO_4^{2-}$) is named "Marine" only because the history of air masses is predominantly oceanic. But this is not mandatory, the oceanic / continental parameter is not considered in the clustering.

The category characterized by high concentrations of $Cl^-$, $Na^+$ and $Mg^{2+}$ is called "Highly marine". This category, with only 31 objects, appear to be modest at PUY (the air masses are mainly transported from the Ocean, 400 km away, to PUY with no relief between). This suggests that some clouds coming from the western sector (which could have been classified as "highly marine") have at least either precipitated or become diluted (increase in liquid water content), thereby decreasing concentration. Then, these western clouds are classified as "marine" with a marine air mass history but presenting low salt concentration.

The AHC also provides two other categories, "continental" and "polluted", with high and the highest concentrations of $NH_4^+$, $NO_3^-$ and $SO_4^{2-}$.

We modified the text as follows:

Line 202:

*"According to the classification proposed by Renard et al. (2020), cloud samples are classified into four categories, "marine", "highly marine", "continental" and "polluted" by means of an agglomerative hierarchical clustering (AHC), based solely on their chemical concentrations ($Cl^-$, $Mg^{2+}$, $Na^+$, $NH_4^+$, $SO_4^{2-}$ and $NO_3^-$). This allows clustering clouds in four categories: "highly marine", "marine", "continental" and "polluted". The "marine" clouds have the lowest ion concentrations and most of them come predominantly from western sectors. Marine category is predominant and the most "homogeneous" in terms of concentrations. The "highly marine" category with a similar air mass history, gathers clouds with the highest sea-salts concentrations ($Cl^-$, $Mg^{2+}$ and $Na^+$). The continental category corresponds mainly to air masses with high concentrations of potentially anthropogenic ions ($NH_4^+$, $NO_3^-$ and $SO_4^{2-}$) arriving predominantly from the northeastern sector. Finally, the "polluted" category gathers cloud samples with the highest anthropogenic ion concentrations. All the data relative to the clouds studied in the present work are reported in Table S1 and come from the PUYCLOUD dataset."*

- Line 255: How is the ABLH determined?

ABLH is the boundary layer height parameter of the ECMWF ERA5 (European Centre for Medium-Range Weather Forecasts Reanalysis v5).

We modified the text in the SI as follows:

*"Figure S4. Individual CAT model back trajectories of each of the 13 cloud events reaching PUY. Colors correspond to the air mass height minus the atmospheric boundary layer height (ABLH). Positive values (> ABLH, red) indicate the air mass is in the free troposphere. Negative values (< ABLH, blue) indicate the air mass is in the boundary layer. ABLH is the boundary layer height parameter of the ECMWF ERA5 (European Centre for Medium-Range Weather Forecasts Reanalysis v5). Its calculation is based on the bulk Richardson number following the conclusions of Seidel et al. (2012). A more detailed description of this parameter and its calculation can be found in the ECMWF operational implementation document (available at: https://confluence.ecmwf.int/display/CKB/ERA5)."*

- Please add information about the volumes of cloud water used for the analysis. Standard addition is a method with many advantages, as the authors pointed out, however it consumed a high amount of samples as several aliquots are needed. Could you please comment on that?

Samples, before LC-MS analysis, are prepared containing the original cloud water, added with 20 AAs at 9 final concentrations (0, 1.0, 5.0, 10, 25, 50, 100, 150 and 500 µg L$^{-1}$). The volume of injection is 5 µL. Hence, around 1 mL is needed. Thus, only few mL of cloud water are consumed for the analysis.

We modified the text as follows:

Line 149:

"*Ten samples ready for LC-MS analysis are prepared, from approximately 1 mL of cloud water, containing the original cloud water added with 20 AAs at final concentrations set to 1.0, 5.0, 10, 25, 50, 100, 150, 500 µg L$^{-1}$.*"

- And did you compare the standard addition to an external or internal calibration to check for possible matrix effects?

In previous experiments when using artificial cloud water composed of different ions at various concentrations, we could clearly see a matrix effect when a simple external calibration was used. It is also well-known that environmental samples represent much complex and variable matrices. In this work, we only used the standard addition to overcome the matrix effect.

The expression "avoid matrix effects" used before is probably excessive.

We modified the text as follows:

Line 14:

"*This quantification has been performed without concentration neither derivatization, using LC-MS and the standard addition method to correct for matrix effects.*"

- Line 299 and following: I am wondering what information can be obtained from this paragraph and Fig. 4. A statistical method is applied but I don´t see a clear new result. It just "confirms" Fig. 3 and this section is not clear to me. Please improve and show the results obtained here.

AHC allows to statically analyze differences in AAs distribution among samples. By only analyzing Figure 3, the preponderance of Ser and Gly suggests that only these two AAs allow to separate the 3 most concentrated samples (1-Oct-18, 25-sep-19 and 2-Oct-19) from the 10 others. Using AHC, we argue that the excess of these two specific AAs does not explain the cloud clusterization in the proposed two groups.

[Figure]

**Figure 3. a. Distribution (nM) and b. relative contributions (% nM) of AAs molar concentrations in each cloud event sampled at PUY.**

We modified the text as follows:

Line 306:

*"Agglomerative hierarchical clustering (AHC), used to categorize cloud samples based on the AA concentration, successfully groups the 13 observations, with a satisfactory cophenetic correlation (correlation coefficient between the dissimilarity and the Euclidean distance matrices) of 0.79/1 (Figure 4a). The dotted line in Figure 4a represents the degree of truncation (dissimilarity = 5.7 $10^6$) of the dendrogram used for creating categories. This truncation is automatically chosen, based on the entropy level. The AHC profile plot (Figure 4b) details the average composition of these two categories determined from the 18 AAs.*

*AHC establish two different categories which reflect the variability of AAs in the 13 cloud samples. In detail, the blue category gathers 10 cloud samples with lower AA concentrations. This blue category is the most homogeneous (within-class variance = 3.7 $10^5$), compared to the red category (within-class variance = 1.2 $10^6$). Conversely, the red one, more heterogeneous, gathers 3 cloud samples with higher AA concentrations except for Met (absent in most of the 13 samples).*

*AHC reveals two categories significantly different, which are not explained by a punctual excess of certain AAs such as Ser of Gly. This cannot be concluded by only analyzing Figure 3 and confirms the gain of using AHC. AHC allows to perform a non-parametric test (Mann-Whitney test, not shown). Because the computed p-value is lower than the significance level alpha = 0.05, the distribution of 9 AAs (Asp, Gly, His, Leu/I, Lys, Phe, Ser, Thr and Tyr) concentrations can be accepted as significantly different between both AHC categories.*

*Note that the 13-Jun-18 and 24-Aug-18 cloud samples are isolated in the AHC blue category due to their high Asn concentration."*

[Figure]

**Figure 4. a. Dendrogram representing the agglomerative hierarchical clustering (AHC) based on dissimilarities using the Ward's method on concentrations of the 18 AAs. The 13 cloud samples are assigned to one of two established categories by entropy (*i.e.*, dissimilarity < 5.7 10⁶). b. Profile plot established by the AHC from the 18 main AAs. The Y axis, in logarithmic scale, displays the average AA concentrations of the category.**

- Line 328: I am in doubt that "analytical tools" can explain the explained discrepancies. Why should different analytical methods show non consistent results on the concentration and composition on the amino acids?

You are right, it is unlikely. We modified as follows:

Line 337:

*"This discrepancy could result from sampling characteristics, i.e., cloud waters in the Bianco's study have been sampled during two short periods (March/April and November 2014), whereas in the present work, cloud waters have been collected over 6 years and covering different seasons."*

- Tab. 2 line "Cabo Verde" HPLC-MS does not go together with OPA 7erivatization. This seems to be an error, please correct. Same for Table S4.

You are right, Triesch et al. (2021) performed derivatization of amino acids using Waters AccQ-Tag precolumn. We modified the text in Tables 2 and S4.

- Table 3: Is "R" or "R^2" shown here? To this: line 392: I am in doubt if a correlation of 0.4 is "significant". If you use this expression, please state your definition of significance. In general, I am not convinced from the shown correlations. Here again, p-values would help.

The PLS regression is a powerful statistical tool that is adapted for particular data conditions such as small sample sizes or data with non-normal distribution *(Chin and Newsted, 1999, page 337)*. However, with only 13 samples, all the results in this work should be considered as "trends" that need to be investigated. p-values are heterogeneous, but overall, below 0.10 for the most significant correlations (*e.g.*, sea surface *vs* AAs).

The text was modified as follows:

Line 406:

*"The PLS regression is a powerful statistical tool adapted for particular data conditions such as small sample sizes or data with non-normal distribution (Chin and Newsted, 1999, page 337). However, with only 13 samples, all the results in this work should be considered as "trends" that need to be investigated."*

Concerning the correlation between the biological parameters and the 9 AAs (Gly, His, Tyr, Asp, Leu/I, Thr, Phe and Ser), with R = 0.4, the adverb "significant" is not suitable and has been removed. Our purpose was to underline that all the trends mentioned (*i.e.*, AA *vs* sea surface, Continental surface (< ABLH), $Na^+$, $K^+$, $Cl^-$, bacteria, and season) relate specifically to these 9 AAs. PLS highlights trends that should be considered as a whole, and the various correlated parameters are gathered within components (t1 or t2) which can be found on the PLS chart (Figure 5). The quality of the modeling of the PLS regression is evaluated by an index: the $Q^2$. The $Q^2$ must be positive for the PLS regression to be predictive. In this PLS (Table 3), with all the parameters, $Q^2$ = 0.044. the predictivity is low, that is why we performed a second PLS limited to the CAT model parameters (Sea and Continental surfaces) as Xs.

The table was modified as follows:

**Table 3. PLS correlation matrix between AA concentrations and "zone matrix" (sea/continental surface </> ABLH) from the CAT model, and temperature, pH, cation and anion concentrations, TOC and H2O2 concentrations, bacteria density (CFU mL$^{-1}$) and ATP concentration, and the seasons (Fall/Winter and Spring/Summer) determined from 13 cloud sampled at PUY. Highest correlations are displayed in dark red and highest anti-correlations in dark blue. R > 0.5 (or R < -0.5) with p_values < 0.1 are underlined.**

| Variables | Ser | Gly | Ala | Asn | Leu/I | Thr | Asp | Pro | Glu | His | Phe | Tyr | Lys | Gln | Arg | Trp | Met |
|---|---|---|---|---|---|---|---|---|---|---|---|---|---|---|---|---|---|
| Sea surface (< ABLH) | **0.88** | 0.68 | 0.18 | **0.38** | **0.76** | **0.82** | **0.74** | 0.31 | **0.53** | **0.70** | **0.84** | **0.71** | **0.58** | 0.21 | 0.20 | 0.08 | 0.08 |
| Sea surface (> ABLH) | **-0.57** | -0.45 | 0.03 | -0.14 | **-0.50** | -0.52 | -0.42 | -0.28 | -0.04 | -0.35 | -0.37 | **-0.58** | -0.11 | 0.16 | 0.31 | 0.38 | 0.25 |
| Continental surface (< ABLH) | **0.54** | 0.33 | 0.45 | 0.26 | 0.57 | 0.63 | 0.55 | **0.63** | 0.33 | 0.56 | 0.22 | **0.68** | 0.18 | 0.17 | -0.48 | -0.06 | 0.00 |
| Continental surface (> ABLH) | -0.31 | -0.21 | -0.26 | -0.23 | -0.27 | -0.32 | -0.33 | -0.11 | -0.48 | -0.36 | -0.40 | -0.18 | -0.43 | -0.36 | -0.38 | -0.41 | -0.31 |
| Temperature (°C) | -0.08 | 0.29 | -0.14 | 0.29 | -0.10 | -0.15 | -0.07 | -0.25 | -0.41 | -0.28 | 0.40 | -0.13 | -0.36 | 0.51 | -0.13 | 0.09 | -0.20 |
| pH | -0.10 | 0.00 | -0.13 | -0.19 | 0.02 | -0.25 | -0.11 | -0.23 | 0.40 | 0.06 | -0.32 | -0.03 | 0.22 | 0.09 | 0.34 | **0.61** | 0.67 |
| Na$^+$ (µM) | **0.77** | **0.93** | 0.17 | -0.07 | 0.62 | 0.44 | **0.65** | 0.03 | 0.07 | 0.46 | **0.80** | 0.54 | 0.43 | 0.02 | 0.03 | 0.12 | -0.01 |
| NH$_4^+$ (µM) | -0.38 | -0.03 | -0.13 | -0.10 | -0.21 | -0.43 | -0.17 | -0.36 | -0.13 | -0.10 | -0.11 | -0.16 | -0.21 | 0.03 | -0.22 | -0.05 | 0.13 |
| Mg$^{2+}$ (µM) | 0.06 | -0.11 | 0.09 | 0.10 | 0.17 | 0.11 | 0.15 | 0.23 | 0.46 | 0.41 | -0.02 | 0.31 | 0.18 | -0.11 | -0.08 | -0.08 | 0.30 |
| K$^+$ (µM) | 0.75 | **0.85** | 0.18 | 0.08 | 0.72 | 0.55 | **0.75** | 0.08 | 0.08 | 0.48 | **0.73** | 0.61 | 0.40 | 0.00 | -0.24 | -0.16 | -0.16 |
| Ca$^{2+}$ (µM) | -0.08 | -0.34 | 0.04 | 0.30 | -0.08 | 0.20 | 0.00 | 0.38 | -0.05 | 0.09 | -0.04 | 0.11 | -0.28 | -0.08 | -0.39 | **-0.53** | -0.41 |
| SO$_4^{2-}$ (µM) | -0.11 | 0.12 | 0.37 | -0.29 | 0.13 | -0.11 | 0.12 | -0.01 | 0.41 | 0.18 | -0.15 | 0.08 | 0.13 | 0.10 | -0.13 | 0.21 | **0.58** |
| NO$_3^-$ (µM) | -0.17 | 0.00 | -0.05 | 0.02 | -0.01 | -0.03 | 0.02 | -0.12 | 0.02 | -0.07 | 0.01 | -0.05 | -0.21 | 0.07 | -0.26 | -0.39 | -0.05 |
| Cl$^-$ (µM) | 0.67 | **0.76** | 0.01 | -0.05 | **0.60** | 0.34 | **0.50** | -0.18 | 0.34 | 0.44 | 0.63 | 0.44 | **0.65** | 0.03 | 0.40 | 0.28 | 0.38 |
| TOC (mgC L$^{-1}$) | 0.03 | 0.09 | -0.09 | 0.25 | 0.00 | -0.10 | -0.10 | 0.15 | -0.35 | -0.09 | 0.16 | 0.19 | -0.26 | 0.04 | -0.27 | 0.06 | -0.16 |
| H$_2$O$_2$ (µM) | -0.19 | -0.01 | -0.44 | -0.10 | -0.26 | -0.46 | -0.40 | -0.32 | -0.26 | -0.25 | -0.24 | -0.15 | -0.16 | 0.00 | 0.21 | 0.43 | 0.20 |
| Bacteria (17°C; CFU/mL) | 0.28 | 0.07 | 0.50 | 0.20 | 0.41 | **0.51** | 0.41 | 0.33 | 0.43 | 0.44 | 0.21 | 0.36 | 0.36 | 0.11 | -0.19 | -0.39 | 0.19 |
| ATP (nM) | 0.13 | **0.58** | 0.00 | -0.14 | 0.19 | -0.04 | 0.22 | -0.32 | 0.05 | 0.19 | 0.33 | 0.17 | 0.01 | 0.19 | -0.16 | 0.14 | 0.16 |
| Fall / Winter | 0.44 | 0.40 | 0.46 | -0.44 | 0.32 | 0.36 | 0.36 | 0.29 | -0.15 | 0.15 | 0.21 | 0.25 | 0.21 | -0.39 | -0.23 | -0.31 | -0.25 |
| Spring / Summer | -0.44 | -0.40 | -0.46 | 0.44 | -0.32 | -0.36 | -0.36 | -0.29 | 0.15 | -0.15 | -0.21 | -0.25 | -0.21 | 0.39 | 0.23 | 0.31 | 0.25 |

- Line 391: " … a potential influence of the photochemistry." on what?

We were referring to photochemical processes capable of degrading AA. To clarify this point, we modified the text as follows:

Line 402:

*"These 9 AAs are also slightly anti-correlated with H$_2$O$_2$ concentration, suggesting a potential influence of the photochemistry on AA concentrations (Lundeen et al., 2014)."*

- Line 394 and Fig. 5, similar comment as above (Fig.4): I am not familiar with this type of graphs and discussion, but I don't see the clear benefit from this chart and discussion. Could you more clearly show the results and its importance?

As explained above, the quality of the modeling of the PLS regression is evaluated by an index: the Q². The Q² must be positive for the PLS regression to be predictive. In the first PLS (Table 3), with all the parameters, Q² = 0.044. the predictivity is low, that is why we performed a second PLS limited to the CAT model parameters (*i.e.*, Sea and Continental surfaces, < > ABLH) as Xs. The Q² is improved, Q² = 0.19, which allows to link air mass history (CAT model) and AA concentrations (*e.g.*, a cloud sample with a high sea surface percentage is likely to have high AA concentrations).

PLS provides equations of the type Y = X W$_h$ C$_h$ + E$_h$, where Y is the matrix of the dependent variables and X is the matrix of the explanatory variables. W$_h$ and C$_h$ are the matrices generated by the PLS algorithm, and E$_h$ is the matrix of the residuals. This equation could be used in the future, subject to improvement, to estimate AA concentration from CAT model data.

To clarify, we modified the text as follows:

Line 408:

*"To go further in modelling the environmental variability of the AA concentrations in our cloud samples, we performed a simplified PLS restricting the Xs to the parameters of the CAT model (i.e., the zone matrix). The predictive quality index of the models obtained with the PLS ($Q^2 = 0.19$ with one component) is satisfactory given the complexity of the cloud composition. In details, Figure 5 displays PLS correlation chart with t component on axes t1 and t2. The main axis (t1) is linked to the ABLH and most of the AAs are correlated with "< ABLH". The t2 axis is linked to the zone (sea / continental surfaces) and it reveals a preponderance of marine influence, which is consistent with the dominant western oceanic air masses at PUY."*

[Figure]

*Figure 5. Partial least squares (PLS) chart with t component on axes t1 and t2. The correlations map superimposes the dependent variables from the chemical matrix (blue circles), the explanatory variables (red diamonds) and the cloud events (green cross).*

*"Cloud water is a complex matrix resulting from the interaction of many factors; cloud samples more influenced by continental zones (northeast) could modify this model, and the predictive model provided by this PLS needs further investigations to be validated. However, it appears that the air mass history remains the prevailing parameter, as observed in Renard et al. (2020), after considering more cloud events. The CAT model could be used to estimate the AA concentration, and this work helps to propose scientific plausible reasons explaining the environmental variability of AA composition."*

- Figure 6 is more "result" that "discussion".

You are right but incorporating these results into the discussion seemed like the best option in terms of balance and clarity.

- Chapter 4.1: I have some difficulties with this section. It seems that some examples are picked to explain the AA composition in relation to biological processes. I wonder how representative these explanations are. More references are needed. For example, the explanation of peptidoglycans as a source strongly needs references. What kind of microorganisms have peptidoglycan in their cell walls? What other types of glycanes exist? In the next chapter you report serine as constituents of diatom. Also here references are missing. I´d recommend to the authors to perform a more thorough discussion of the possible biogenic connection rather than choosing some examples that seem to fit to their results. What about long range transport of glycine? What about other serine sources (besides diatom)? I believe there are also more references on the composition of amino acids in seawater during a bloom period besides the one from Ittekkot, 1982. Biological processes are certainly one source of amino acids, but this discussion does not seem to improve the current knowledge at its current state.

As required by the referee we have expanded our literature search on the relative distributions of AA in proteins extracted from different taxa (archaea, bacteria, and eukaryotes) as well as in peptidoglycans. We found that this distribution was rather close to that observed in our cloud samples. We have therefore modified the introduction of section 4.1 and added some references, while keeping the examples concerning serine which is the dominant AA in this work.

We modified the text as follows:

Line 437:

*"Studies report the relative distributions of AAs in proteins extracted from different taxa (archaea, bacteria and eucaryotes) (Bogatyreva et al., 2006; Gaur, 2014; Jordan et al., 2005). Although there are some differences between mammalian, invertebrate, plant, protozoa, fungi and bacterial protein composition, some AAs (Ala, Gly, Leu/I and Val) are clearly dominant while others are in low amounts (Cyst, Trp, His and Met). Globally, this relative abundance of AAs initially constituting proteins presents similarities with the relative concentrations present in our samples. In particular, Gly, Ala, and Leu/I are the most abundant in our samples as in the proteins, while Trp, and Met which concentrations are the lowest in our cloud samples are also minor components of proteins. Ser which is the major AA in our samples is present in proteins in average and is not dominant.*

*We looked at the composition of peptidoglycan that form all the cell wall of Gram positive and Gram negative bacteria that can be an atmospheric source of AAs (Vollmer et al., 2008). Peptidoglycans are complex structures formed by glycan strands (composed of sugars) cross-linked by short penta-peptides. Although some slight variations can exist depending on the bacterial strains, the standard sequence of this peptide is L-Ala-D-Glu-L-Lys-D-Ala-D-Ala. In a few cases Ser and Gly have been also reported in the sequence. In addition, these penta-peptides are connected by inter-peptide bridges varying from 1 to 7 AAs which contain in majority Gly and Ala but also Orn, Lys, Glu or Ser. Peptidoglycans can thus represent a major source of Ala and Gly, that are the major AAs detected in our samples.*

*Finally, we specifically searched in aqueous media for the potential origin of Ser, which is dominant in our sample. Hecky et al. report the AA composition of cell walls from six different diatom species, selected on the basis of taxonomy and habitat diversity (Hecky et al., 1973). Three are from estuarine origin, the other from fresh waters. The protein template of these cell walls is composed of the following AA sequences: Asp-Ser-Ser-Gly-Thr-Ser-Ser-Asp-Ser-Gly. Ser is thus highly abundant in these aquatic organisms and plays an important role in the complexation of the silicon ($Si4+$). This result confirms previous reported data who showed the prevalence of serine in marine diatoms (Chuecas and Riley, 1969). AAs in sea water during phytoplankton blooms were also investigated (Ittekkot, 1982): Glu concentration is maximum in the early stages of the bloom, while Asp, Gly, Ala and Lys concentrations increase at the end of the bloom. In parallel, Ser was one of the most abundant AA and its concentration remains high all along the bloom period. Ser could come from the cell walls of some phytoplankton species which are diatoms. Hashioka et al. showed that diatoms could contribute up to 80% of the total phytoplankton in the ocean to during bloom events (Hashioka et al., 2013). The high Ser concentration measured in our cloud samples could thus originate from diatoms and could be a marker of their oceanic origin; this has also been proposed by Triesch et al. (2021) who underline the marine origin of Ser present in their samples.*

*In conclusion, combining the composition of proteins, peptidoglycans and diatom cell-walls shows that Ala, Gly, Ser, Leu/l are major AAs, while Trp and Met are minor ones; these ratios are fitting rather well with the concentrations found in our cloud samples."*

- "Biological processes are certainly one source of amino acids, but this discussion does not seem to improve the current knowledge at its current state."

Here we do not discuss the biotransformation of AAs but the real source which is for sure from biological origin (proteins, cell walls…). Usually, looking at the composition of these components is not taken into consideration even if they also explain part of our results. Abiotic transformations are important but there are also dependent on the initial amount of AAs present in aerosols or clouds.

- Line 495: specify "one".

Line 537:

*"These theoretical lifetimes of AAs could explain the very low Met and Trp concentrations measured in our cloud samples which are very reactive, and the high concentrations measured for Gly and Asn, and in some extent for Ser and Ala, which are very slowly transformed."*

- Line 524/524: It is not clear if you are referring to the present study or to a result from literature.

Table 4 displayed theoretical lifetimes, columns A, B and C (Triesch et al., 2021; this study adapted from Jaber et al., 2021; McGregor and Anastasio, 2001), and experimental lifetimes, columns D, E and F. (this study adapted from Jaber et al., 2021; McGregor and Anastasio, 2001).

We modified the text as follows:

Line 549:

*"The experimental results displayed Table 4 (Columns D, E and F), which are different from theoretical ones (Columns A, B and C), reflect a much higher complexity of the occurring transformations."*

- Line 539: same comment as above. If you refer to a literature study please add a reference.

We modified the text to clarify:

Line 566:

*"Potential biological transformation processes have been also evaluated in the lab. Recent works (Xu et al., 2020; Zhu et al., 2021), based on Degradation Index (DI) calculations, suggest that biodegradation of AAs could occur in rain and in aerosols."*

- Line 553: please replace or specify "behavior"

We modified the text to clarify:

Line 580:

*"The complexity comes from how different microorganisms use these pathways. Up to now the only biodegradation rates related to atmospheric waters are from Jaber et al. (2021) and might be incomplete; more experimental work should be conducted on real atmospheric samples."*

- Line 563: There seems to be something missing at the end of the sentence (AAs concentration/composition?)

We modified as follows:

Line 587:

*"These AA relative proportions can be explained by the original biological matrices that emit AAs into the atmosphere, by the hygroscopicity of AAs that favors their incorporation in the cloud water and finally by their transformations during their transport into the atmosphere that modulate the total concentration of AAs."*

- Tables in the SI: I strongly recommend to insert the actual LOQ (rather that stating "below LOQ"). The authors apply a method without any enrichment/preconcentration steps, therefore it is important to see the sensitivity of the method right away. It took me a while to find the values in the manuscript.

We explained, in section 3.1, the standard deviation of the measurements ($STD_M$) values, as calculated in this work in the context of a standard addition (equation detailed in Figure S3), could be compared to the limit of quantification (LOQ) established in works using internal standard method (Bader, 1980). We added the $STD_M$ values in the Table S3.

Line 236:

*"The $STD_M$ values, as calculated in this work in the context of a standard addition (equation detailed in Figure S3), could be compared to the limit of quantification (LOQ) established in works using internal standard method (Bader, 1980). Both equations are similar and provide comparable results. However, the precision depends on the number of standard points added in the method, and not on the number of replicates. The values in this work are globally low and consistent with those reported in previous works on cloud waters and aerosol particles (Table S4). A recent study performed by Triesch et al. (2021) was able to quantify Val in cloud water samples, but they could not measure Arg, Asn, His, Lys, Cys and Tyr concentrations. Triesch et al. (2021) is also based on LC-MS (Orbitrap™), but with samples concentrated (factor 44) and derivatized with a pre-column. They reported LOQ values ranging from 0.2 to 1.0 µg $L^{-1}$, vs $STD_M$ from 1.1 to 4.6 µg $L^{-1}$ in this study. $STD_M$ values are also within the same range of magnitude than those*

*reported on aerosol particles by Helin et al. (2017) using direct injection of extracted AAs in LC-MS (triple-quadrupole technology), with values varying from 4 to 160 nM, vs $STD_M$ values from 8 to 44 nM in this work."*

- Why are several samples classified as "marine" (Tab. S1, page 2) although they spent a lot of time over continental surfaces (Tab. S1, page8), for example: the "13 Jun" sample?

As explained before (*Line 200: Why do "marine" clouds have a low ion concentration? I'd expect some levels of sea salt*), the four categories, "Marine", "Highly marine", "Continental" and "Polluted" are established by means of an agglomerative hierarchical clustering (AHC), based solely on their chemical concentrations ($Cl^-$, $Mg^{2+}$, $Na^+$, $NH_4^+$, $SO_4^{2-}$ and $NO_3^-$). The category with low ion concentrations ($Cl^-$, $Mg^{2+}$, $Na^+$, $NH_4^+$, $NO_3^-$ and $SO_4^{2-}$) is named "Marine" only because the history of air masses is predominantly oceanic. But this is not mandatory, the oceanic / continental parameter is not considered in the clustering. Some "Marine" samples may have spent lot of time over the continent and present low ionic concentrations. The "marine" category is the main category (113 objects) at PUY. Within this category, the differences between samples can be significant. Ion concentrations are only considered low compared to the rest of the 300 samples.

**References**

Bader, M.: A systematic approach to standard addition methods in instrumental analysis, J. Chem. Educ., 57, 703, https://doi.org/10.1021/ed057p703, 1980.

Bogatyreva, N. S., Finkelstein, A. V., and Galzitskaya, O. V.: Trend of amino acid composition of proteins of different taxa, J. Bioinform. Comput. Biol., 04, 597–608, https://doi.org/10.1142/S0219720006002016, 2006.

Chin, W. and Newsted, P.: Structural Equation Modeling Analysis with Small Samples Using Partial Least Square, Statistical Strategies for Small Sample Research, 1999.

Chuecas, L. and Riley, J. P.: The Component Combined Amino Acids Of Some Marine Diatoms, J. Mar. Biol. Ass., 49, 117–120, https://doi.org/10.1017/S0025315400046440, 1969.

Deguillaume, L., Charbouillot, T., Joly, M., Vaïtilingom, M., Parazols, M., Marinoni, A., Amato, P., Delort, A.-M., Vinatier, V., Flossmann, A., Chaumerliac, N., Pichon, J. M., Houdier, S., Laj, P., Sellegri, K., Colomb, A., Brigante, M., and Mailhot, G.: Classification of clouds sampled at the puy de Dôme (France) based on 10 yr of monitoring of their physicochemical properties, Atmos. Chem. Phys., 14, 1485–1506, https://doi.org/10.5194/acp-14-1485-2014, 2014.

Gaur, R. K.: Amino Acid Frequency Distribution Among Eukaryotic Proteins, IIOAB journal, 5, 6, 2014.

Hashioka, T., Vogt, M., Yamanaka, Y., Le Quéré, C., Buitenhuis, E. T., Aita, M. N., Alvain, S., Bopp, L., Hirata, T., Lima, I., Sailley, S., and Doney, S. C.: Phytoplankton competition during the spring bloom in four plankton functional type models, 10, 6833–6850, https://doi.org/10.5194/bg-10-6833-2013, 2013.

Hecky, R. E., Mopper, K., Kilham, P., and Degens, E. T.: The amino acid and sugar composition of diatom cell-walls, Marine Biology, 19, 323–331, https://doi.org/10.1007/BF00348902, 1973.

Helin, A., Sietiö, O.-M., Heinonsalo, J., Bäck, J., Riekkola, M.-L., and Parshintsev, J.: Characterization of free amino acids, bacteria and fungi in size-segregated atmospheric aerosols in boreal forest: seasonal patterns, abundances and size distributions, 17, 13089–13101, https://doi.org/10.5194/acp-17-13089-2017, 2017.

Ittekkot, V.: Variations of dissolved organic matter during a plankton bloom: qualitative aspects, based on sugar and amino acid analyses, Marine Chemistry, 11, 143–158, https://doi.org/10.1016/0304-4203(82)90038-X, 1982.

Jaber, S., Joly, M., Brissy, M., Leremboure, M., Khaled, A., Ervens, B., and Delort, A.-M.: Biotic and abiotic transformation of amino acids in cloud water: experimental studies and atmospheric implications, 18, 1067–1080, https://doi.org/10.5194/bg-18-1067-2021, 2021.

Jordan, I. K., Kondrashov, F. A., Adzhubei, I. A., Wolf, Y. I., Koonin, E. V., Kondrashov, A. S., and Sunyaev, S.: A universal trend of amino acid gain and loss in protein evolution, 433, 633–638, https://doi.org/10.1038/nature03306, 2005.

Lundeen, R. A., Janssen, E. M.-L., Chu, C., and Mcneill, K.: Environmental Photochemistry of Amino Acids, Peptides and Proteins, CHIMIA, 68, 812–817, https://doi.org/10.2533/chimia.2014.812, 2014.

Renard, P., Bianco, A., Baray, J.-L., Bridoux, M., Delort, A.-M., and Deguillaume, L.: Classification of Clouds Sampled at the Puy de Dôme Station (France) Based on Chemical Measurements and Air Mass History Matrices, Atmosphere, 11, 732, https://doi.org/10.3390/atmos11070732, 2020.

Seidel, D. J., Zhang, Y., Beljaars, A., Golaz, J.-C., Jacobson, A. R., and Medeiros, B.: Climatology of the planetary boundary layer over the continental United States and Europe, 117, https://doi.org/10.1029/2012JD018143, 2012.

Triesch, N., van Pinxteren, M., Engel, A., and Herrmann, H.: Concerted measurements of free amino acids at the Cabo Verde islands: high enrichments in submicron sea spray aerosol particles and cloud droplets, Atmos. Chem. Phys., 21, 163–181, https://doi.org/10.5194/acp-21-163-2021, 2021.

Vollmer, W., Blanot, D., and De Pedro, M. A.: Peptidoglycan structure and architecture, FEMS Microbiol Rev, 32, 149–167, https://doi.org/10.1111/j.1574-6976.2007.00094.x, 2008.

Xu, Y., Xiao, H., Wu, D., and Long, C.: Abiotic and Biological Degradation of Atmospheric Proteinaceous Matter Can Contribute Significantly to Dissolved Amino Acids in Wet Deposition, Environ. Sci. Technol., 54, 6551–6561, https://doi.org/10.1021/acs.est.0c00421, 2020.

Zhu, R.-G., Xiao, H.-Y., Luo, L., Xiao, H., Wen, Z., Zhu, Y., Fang, X., Pan, Y., and Chen, Z.: Measurement report: Hydrolyzed amino acids in fine and coarse atmospheric aerosol in Nanchang, China: concentrations, compositions, sources and possible bacterial degradation state, 21, 2585–2600, https://doi.org/10.5194/acp-21-2585-2021, 2021.

---

## Author Response (AR2)

Clermont-Ferrand (France), 10th of January 2022

Dear Editor,

    Please find our manuscript "Free amino acids quantification in cloud water at the puy de Dôme station (France)" by Pascal Renard, Maxence Brissy, Florent Rossi, Martin Leremboure, Saly Jaber, Jean-Luc Baray, Angelica Bianco, Anne-Marie Delort and Laurent Deguillaume.

    After discussion with the editorial support of ACP, we send you the final version of the manuscript that need English correction (as suggested by reviewer 1) by your copy-editing service.

Yours Sincerely,

Dr. A.-M. Delort       Dr. Laurent Deguillaume

ICCF            LaMP/OPGC

Université Clermont Auvergne

63178 Aubière Cedex, France

**E-mail: a-marie.delort@uca.fr & laurent.deguillaume@uca.fr**